# Minutes-timescale 3D isotropic imaging of entire organs at subcellular resolution by content-aware compressed-sensing light-sheet microscopy

Chunyu Fang[1,5], Tingting Yu[2,3,5], Tingting Chu[1], Wenyang Feng[1], Fang Zhao[1], Xuechun Wang[1], Yujie Huang[4], Yusha Li[2], Peng Wan[2], Wei Mei[4 ✉], Dan Zhu [2,3 ✉] & Peng Fei [1 ✉]

Rapid 3D imaging of entire organs and organisms at cellular resolution is a recurring challenge in life science. Here we report on a computational light-sheet microscopy able to achieve minute-timescale high-resolution mapping of entire macro-scale organs. Through combining a dual-side confocally-scanned Bessel light-sheet illumination which provides thinner-and-wider optical sectioning of deep tissues, with a content-aware compressed sensing (CACS) computation pipeline which further improves the contrast and resolution based on a single acquisition, our approach yields 3D images with high, isotropic spatial resolution and rapid acquisition over two-order-of-magnitude faster than conventional 3D microscopy implementations. We demonstrate the imaging of whole brain (~400 mm$^3$), entire gastrocnemius and tibialis muscles (~200 mm$^3$) of mouse at ultra-high throughput of 5~10 min per sample and post-improved subcellular resolution of ~1.5 μm (0.5-μm iso-voxel size). Various system-level cellular analyses, such as mapping cell populations at different brain sub-regions, tracing long-distance projection neurons over the entire brain, and calculating neuromuscular junction occupancy across whole muscle, are also readily accomplished by our method.

[1] School of Optical and Electronic Information- Wuhan National Laboratory for Optoelectronics, Huazhong University of Science and Technology, 430074 Wuhan, China. [2] Britton Chance center for Biomedical Photonics, Wuhan National Laboratory for Optoelectronics, Huazhong University of Science and Technology, 430074 Wuhan, China. [3] MoE Key Laboratory for Biomedical Photonics, Huazhong University of Science and Technology, 430074 Wuhan, China. [4] Department of Anesthesiology, Tongji Hospital, Tongji Medical College, Huazhong University of Science and Technology, 430030 Wuhan, China. [5] These authors contributed equally: Chunyu Fang, Tingting Yu. ✉email: wmei@hust.edu.cn; dawnzh@hust.edu.cn; feipeng@hust.edu.cn

The comprehensive understanding of majority and singular cellular events, as well as their complex connections in whole organs and organisms, is one of the fundamental quests in biology. To extract the various cellular profiles of different physiological functions within specimens, three-dimensional (3D) high-resolution imaging is required throughout a mesoscale-sized volume. However, creating such a large-scale image dataset has posed significant challenges for conventional 3D light microscopy methods, which suffer from relatively small optical throughputs, known as the amount of spatial information provided per unit time[1,2], and limited imaging depth (up to a few hundred of microns) unable to extract signals from deep tissues. A common strategy for addressing these issues has become to use 3D tile stitching combined with tissue sectioning[2–5]. For example, both tiling confocal microscopy[6–9] and sequential two-photon tomography[10–13] (STPT) can three-dimensionally image mouse brain at subcellular resolution, but do so at the expense of a long acquisition time, due to the slow laser-point-scanning and extra time consumption on mechanical stitching and slicing. The recent integration of light-sheet microscopy[14–17] and tissue clearing[18–21] has facilitated an important alternative to conventional 3D histological imaging approaches by instead applying a nondestructive light-sheet to selectively illuminate a thin plane of the clarified sample. At the same time, the use of wide-field detection results in improved imaging speed as compared to the epi-fluorescence approaches. Although light-sheet microscopy using a Gaussian-type hyperbolic light-sheet[2,15,22] has been successfully used for imaging large organ samples, a tradeoff exists between the minimum thickness of the light-sheet and the confocal range over which it remains reasonably uniform. As a reference, when illuminating a range of 1 mm in length, an optimized Gaussian light-sheet diverges to a full-width at half-maximum (FWHM) thickness of 7.5 μm at either end, and is hence incapable of resolving fine neuronal fibers across a large volume of brain. Sweeping the waist of Gaussian light-sheet (axially swept light-sheet microscopy, ASLM) along the propagation direction can extend the imaging field of view (FOV) while keep a thin light-sheet thickness, thereby greatly improving the axial resolution[23–25]. The recent mesoSPIM has achieved 5.5-μm axial resolution with 3.3-mm FOV[24], while ctASLM has demonstrated 0.95-μm resolution under 737-μm FOV[25]. However, like confocal microscopy, excessive out-of-waist excitation by ASLM approaches also causes nonnegligible photobleaching to samples[26]. Besides sweeping the waist of Gaussian beam to obtain a thin-and-wide plane illumination, Bessel-type light-sheet microscopy[27–29] (LSM) can also generate a non-diverging light-sheet with large FOV and high axial resolution. Unlike ASLM, Bessel LSM can retain most of the beam energy in its central lobe at focal plane so that the photobleaching is relatively low. Meanwhile, to minimize the effect from side-lobe excitation[30], Bessel LSM usually uses a high numerical-aperture (NA) detection objective with a small depth-of-focus. Thus, it is mainly optimized for the live imaging of organelles in a single or a few cells. Furthermore, as with epifluorescence methods that suffer from a tradeoff between accuracy and scale, either a Gaussian or a Bessel LSM system still has limited optical throughput, highlighting the difficulty for obtaining high spatial resolution across a very large FOV[31]. To image a whole mouse brain at subcellular resolution, a small step size and tile stitching at high magnification are necessary to acquire trillions of sub-μm³ voxels over a cm³-sized volume[2,19]. Therefore, a long acquisition time from several hours to days, as well as increased photobleaching to samples, still prevents the widespread applications of LSM to the high-resolution mapping of entire mammalian organs/organisms. Compressed-sensing (CS) computation, known as its ability to recover higher-quality signal from single incomplete measurement, is a solution to this throughput issue[32]. Conventional CS has been previously implemented on light microscopy to reduce the acquisition time via either recovering degraded signals from incomplete measurements, or increased the signal-to-noise ratio (SNR) of acquired images[33,34]. However, regular CS recovery is sensitive to the signal characteristics, such as the sparsity and noise level. Thus, its application to large-scale 3D image, in which the signal sparsity and dynamic range may vary drastically within a large volume[2], remains highly challenging. While approach like nonlocal total variation[35] and Laplacian scale mixture model[36] can adaptively regularize CS computation according to the content feedbacks from a few sub-regions of a photograph, it shows limited performance of restoring the decimated signal details.

Here we show that combining a large-FOV scanning Bessel light-sheet microscopy with a 3D content-aware compressed-sensing (CACS) computation procedure, can address the above-mentioned imaging challenges. A dual-side confocally-scanned Bessel light-sheet with a millimeter-to-centimeter tunable range is developed to illuminate regions-of-interest (ROIs) from a few cells to entire mouse organs, providing uniform optical sectioning of deep tissues with 1–5 μm ultra-thin axial confinement. In addition to the optical design, our content-aware compressed-sensing (CACS) procedure can automatically extract inconstant signal characteristics for different sample regions, and appropriately regularize the CS computation to further improve the quality of the acquired 3D image based on a single input of its own, allowing satisfying spatial resolution with data acquisition shorten by two-orders of magnitude. Our method thus overcomes the limitations of anisotropic resolution as well as inadequate throughput from current whole-organ imaging techniques. We apply this method to the high-throughput anatomical imaging of whole mouse brain to demonstrate its unique capabilities, such as the rapid screening of multiple brains in minutes, instant imaging of any region of interest in selected brain at subcellular resolution in seconds, and teravoxel high-resolution mapping of entire brain of ~400 mm³ on a timescale of ~10 min. Such a readily-accessible whole-brain dataset allows us to perform various system-level cellular profiling, such as segmentation of brain regions, cell population counting, and tracing of neural projections. Furthermore, we also demonstrate dual-color imaging of subcellular neuromuscular junctions across an ~200 mm³ volume of entire mouse muscle in ~5 min, thus enabling efficient quantification of the neuromuscular junction occupancy at whole tissue scale.

## Results

### Zoomable, line-synchronized, CACS Bessel plane illumination microscopy.

The optical layout and photographs of our self-built Bessel light-sheet microscope are shown in Supplementary Fig. 1. The simple mode of operation involves sweeping the Bessel beam in the y direction to create a continuous sheet at each z plane. However, in this mode, the cross-sectional profile of the excitation sheet contains broad tails because of the combined influence of the side lobes[28] (Supplementary Fig. 3). Given the fact that the increase in depth-of-field is proportional to the square of the decrease in NA, the axial excitation from these side lobes is more likely to deteriorate the fluorescence detection in our low-to-middle magnification/NA setup (Supplementary Fig. 4). A confocal slit was therefore formed by tightly synchronizing the sweeping Bessel beam with the rolling active pixel lines of the camera[37,38], to block the influence of residing fluorescence excited by the side lobes, and led to much less background (Supplementary Figs. 3 and 4, Movie S1). As a result, the microscopy system could produce sharp optical sectioning at a very large scale, for example, generating a 4 × 4 mm illuminating light-sheet with a cross-sectional profile of ~1.6 μm in thickness (FWHM), as compared with a ~15-μm conventional Gaussian sheet covering the same FOV (Supplementary Fig. 3, Movie 1). When this wide-and-thin Bessel plane illumination combined with a matched magnification (×3.2) was applied to the 3D imaging of cleared mouse

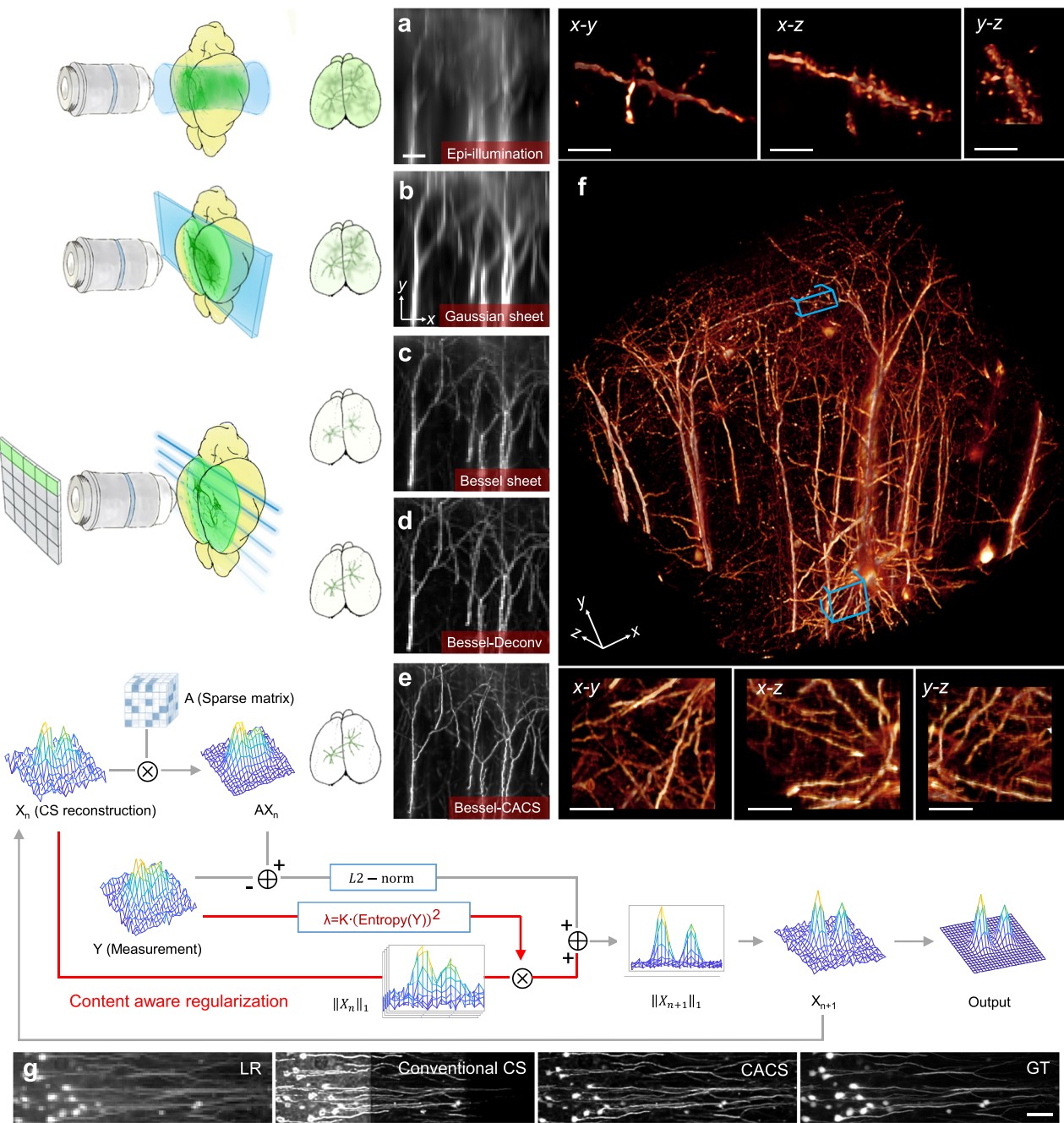

**Fig. 1 CACS Bessel plane illumination microscopy and other imaging modes. a** Epi-illumination mode (left), and maximum-intensity-projection (MIP) in the *y-z* plane from a 3D image stack of a cleared transgenic mouse brain tissue with GFP labelled to the neurons (right, ×3.2/0.28 objective). **b** Gaussian sheet mode (left) showing plane illumination at selective depth of brain, and *y-z* plane MIP from the same sample (right, 0.02 NA illumination + ×3.2/0.28 detection). **c–e** Raw, Deconvolution and CACS modes under our synchronized Bessel plane illumination geometry (0.14 NA illumination + ×3.2/0.28 detection), showing much sharper optical sectioning (**c, d** left) and how CACS computation being further applied to super-resolve the image (**e** left). The *y-z* plane MIPs by each mode are accordingly shown in right columns. Scale bar, 20 μm. **f** Volume rendering of GFP-labelled neurons in a cleared mouse cortex tissue by ×3.2 Bessel-CACS mode. Insets: MIPs along orthogonal axes of the cubical volume of interest (blue boxes). Scale bar, 10 μm.
**g** Comparison between raw LR image, conventional CS recovery with constant parameters, CACS recovery with adaptive parameters, and ×12.6 ground-truth image. Each experiment was repeated three times independently with similar results. Scale bar, 100 μm.

brain tissue with neurons labelled by green fluorescent protein (Thy1-GFP-M), the maximum-intensity projection (MIP) in the *y-z* plane of the cortical area (Fig. 1c) showed axial resolution and contrast superior to those from ×3.2 epi-illumination and ×3.2 Gaussian plane illumination (Fig. 1a, b).

The raw Bessel sheet working at such a relatively large FOV permits rapid 3D imaging of whole brain without many times of

stitching, but may still show inadequate resolutions, e.g., ~4.5-μm isotropic resolution under ×3.2 plus 2-μm *z* step size remaining inadequate for tracing neuronal fibers. Normally, the raw Bessel sheet data can be deconvolved with appropriate point spread functions (PSFs), to slightly improve image quality (Fig. 1d). In our Bessel light-sheet setup, the camera's under-sampling in the *x-y* plane and sparse *z*-excitation by thin plane illumination leads

to a relatively sparse distribution of signals in three dimensions, which allows application of compressed sensing. To adopt this approach to our large-scale Bessel imaging, (Supplementary Figs. 6–9), we extended the CS computation to three dimensions and applied a content-aware (CA) regularization, which was highly adaptive to the varying signal characteristics at different sample regions (Fig. 1e). Using our CACS processing, we aim to achieve a higher-quality image $X$ from the raw large-scale Bessel image $Y$, which shows inadequate resolution resulting from the voxelization and limited numerical aperture. The computation procedure was implemented in Fourier domain, in which the spatial signals could be more sparsely represented. We first obtained the compressed matrix $A$ in Fourier domain based on the system PSF, and applied it to the conventional CS computation term. The entire Bessel image stack $Y$ was divided into many small volumes and transformed into Fourier representations $y_i$, to calculate their specific entropy parameter $\beta_i$ that represented the degree of signal disorder and density parameter $\alpha_i$ that indicated the signal density, for each $y_i$ (Supplementary Note 3). A regularization factor $\lambda_i$ varying with the image content in each small volume was then determined by the product of $\alpha_i$ and $\beta_i$, and applied to the weighting of a regularization term $x_{i1}$, which was the $L1$-norm of the Fourier representations of high-resolution $x_i$ to be solved (Supplementary Note 3). This content-aware regularization procedure with appropriate $\lambda_i$ balances the iterative signal reconstruction process between an overfitting result subject to excessive constraints from complicated signals and an under-fitting result with too sparse signals accompanied by obvious artefacts that are difficult to be further optimized by iteration (Fig. 1g, Supplementary Note Fig. 1, Movie 2). A higher-resolution image tile $x_i$ can be recovered by finding the optimal solution of the following $L1$-regularized least squares problems in the sparse Fourier domain:

$$\text{minimize} \|Ax_i - y_i\|_2^2 + \lambda_i \|x_i\|_1 \qquad (1)$$

With multiple $x_i$ being solved in parallel, they were transformed back to spatial volumes and stitched together to generate the final high-resolution large-scale output $X$ (Fig. 1g, Supplementary Note Fig. 1, Movie 2). Figure 1e shows the super-resolved $y$-$z$ MIP of the neurons originally by ×3.2 Bessel sheet, after a four-times CACS computation applied. Figure 1f shows a CACS-reconstructed volume rendering of apical dendrites in a region of the cortex yielding sufficiently high isotropic 3D resolution and signal-to-noise ratio (SNR) to discern subcellular structures such as dendrite spines. This content-aware regularization is especially suited for the recovery of large-scale image, in which the signals property changes drastically. Conventional CS implementation with fixed regularization causes obvious artefacts and signal-loss in this case, while the CACS can recover relatively accurate signals with substantial resolution improvement (Fig. 1g, Supplementary Note Fig. 1). Besides the recovery of signals originally imaged at relatively low magnification, CACS procedure was also applicable to higher magnifications, such as ×12.6, for resolving ultra-fine dendrite spines at deep subcellular resolution (Supplementary Fig. 10).

**Comparisons with SPIM, confocal, and TPEM.** As confocal microscopy, two-photon-excitation microscopy (TPEM) and selective plane illumination microscopy (SPIM) are the current workhorses for 3D imaging of large organs, we compared them with various modes of Bessel beam plane illumination. Image slices in the $x$-$z$ plane from dendrites of Thy1-GFP-M mouse brain demonstrating comparative axial resolutions were acquired using a laser-scanning epifluorescence microscope under confocal and two-photon-excitation (TPE) modes (Nikon Ni-E, CFI LWD ×16/0.8W

objective), a SPIM (0.02 NA illumination, ×3.2/0.28 detection objective), the low-resolution Bessel sheet mode (0.14 NA illumination + ×3.2/0.28 detection), the high-resolution Bessel sheet mode (0.28 NA illumination + ×12.6/0.5 detection objective) and the CACS-enabled Bessel sheet mode (Fig. 2a–f). With the confocal and TPE microscopes, the anisotropic PSFs in the epifluorescence mode influenced the visualization of the dendrite fibers in the longitudinal direction (Fig. 2a, b). The thick Gaussian sheet covering the full imaging FOV (~4 mm) also gave insufficient axial resolution (FWHM ~15 μm, Fig. 2c), which was obviously poorer than that of the other methods. The raw ×3.2 Bessel sheet covering the same FOV gave higher axial resolution superior to the Gaussian sheet (Fig. 2d). In contrast, the ×12.6 Bessel sheet mode and CACS reconstruction both demonstrated a clear reduction in out-of-focus haze, as well as high axial resolution superior to the Gaussian sheet, ×3.2 Bessel sheet, confocal and TPE methods (Fig. 2e, f). The CACS reconstruction substantially recovered the ultrastructures of the dendrites (Fig. 2f), which remained unresolvable in the raw ×3.2 Bessel sheet image (Fig. 2d). In the meantime, the recovery fidelity was shown to be sufficiently high, as compared to the ×12.6 Bessel sheet image (Fig. 2g, h). The linecuts through individual dendrite fibers (Fig. 2a–f) made by each method further reveal the narrower axial and lateral FWHMs of the CACS Bessel sheet modes in comparison with the regular Bessel sheet. The experimentally measured axial FWHMs were around 15, 4.5, 1.5, 1.5, 4.2, and 4 μm for Gaussian sheet, ×3.2 Bessel sheet, ×12.6 Bessel sheet, ×3.2 CACS Bessel sheet, confocal, and TPE methods, respectively (Fig. 2i). The corresponding lateral values were around 4.5, 4.5, 1.5, 1.5, 0.85, and 0.85 μm. It is noted that to reduce the acquisition time, the $z$-scan step size was herein set to 2 μm for ×3.2 Bessel sheet. Owing to the under-sampling, the measured axial FWHM of dendrites is larger than the native Bessel sheet thickness (~1.6 μm) measured via PSF test (0.5-μm step size, Supplementary Fig. 3). Aside from the resolution improvement, the CACS procedure also significantly improved the SNR, as shown in Supplementary Fig. 12.

Large-scale imaging often needs to balance the image quality with the acquisition time, owing to the insufficient throughput fundamentally limited by the space bandwidth product (SBP) of the system, which is defined as (volumetric imaging speed)/(achievable spatial resolution)[31]. Therefore, precise measurements in a reasonably short acquisition time with a low photobleaching rate remains a big challenge. For example, confocal and TPE microscopes can three-dimensionally image a sample at fairly high resolution (3 μm³), but do so at the cost of low speed due to the point-scanning detection ($4.2 \times 10^4$ μm³/s), and noticeable phototoxicity caused by the epi-illumination mode. In contrast, Gaussian or Bessel sheet modes with in-plane excitation and camera-based detection can image the sample at video rate speed and with lower photobleaching. The CACS-enabled Bessel sheet mode further improves the volumetric resolution by about 30 times, while maintaining the same acquisition speed and bleaching rate, thereby resulting in a much higher throughput.

We compared the degree of photobleaching of the five imaging methods by repeatedly imaging the cortex area of a Thy1-GFP-M mouse brain 30 times (Supplementary Fig. 13). After normalizing for differences in SNR (see Methods), we calculated the averaged signal intensity of each volume for each method, and plotted their variations in Fig. 2j. After 30 cycles of imaging of the same volume, ~95%, 90%, 80%, 40%, and 30% of the fluorophores were preserved by the Gaussian sheet, ×3.2 Bessel sheet, ×12.6 Bessel sheet, TPE, and confocal methods, respectively[1,21]. In Fig. 2k and Supplementary Note Table 1, we further rate the overall imaging performance of confocal, TPE, Gaussian sheet, ×3.2 Bessel sheet, ×12.6 Bessel sheet and CACS Bessel sheet methods by comparing their volumetric resolutions, acquisition speeds, and photobleaching rates. Under the magnification of ×3.2 used for whole-brain imaging, the CACS Bessel sheet mode

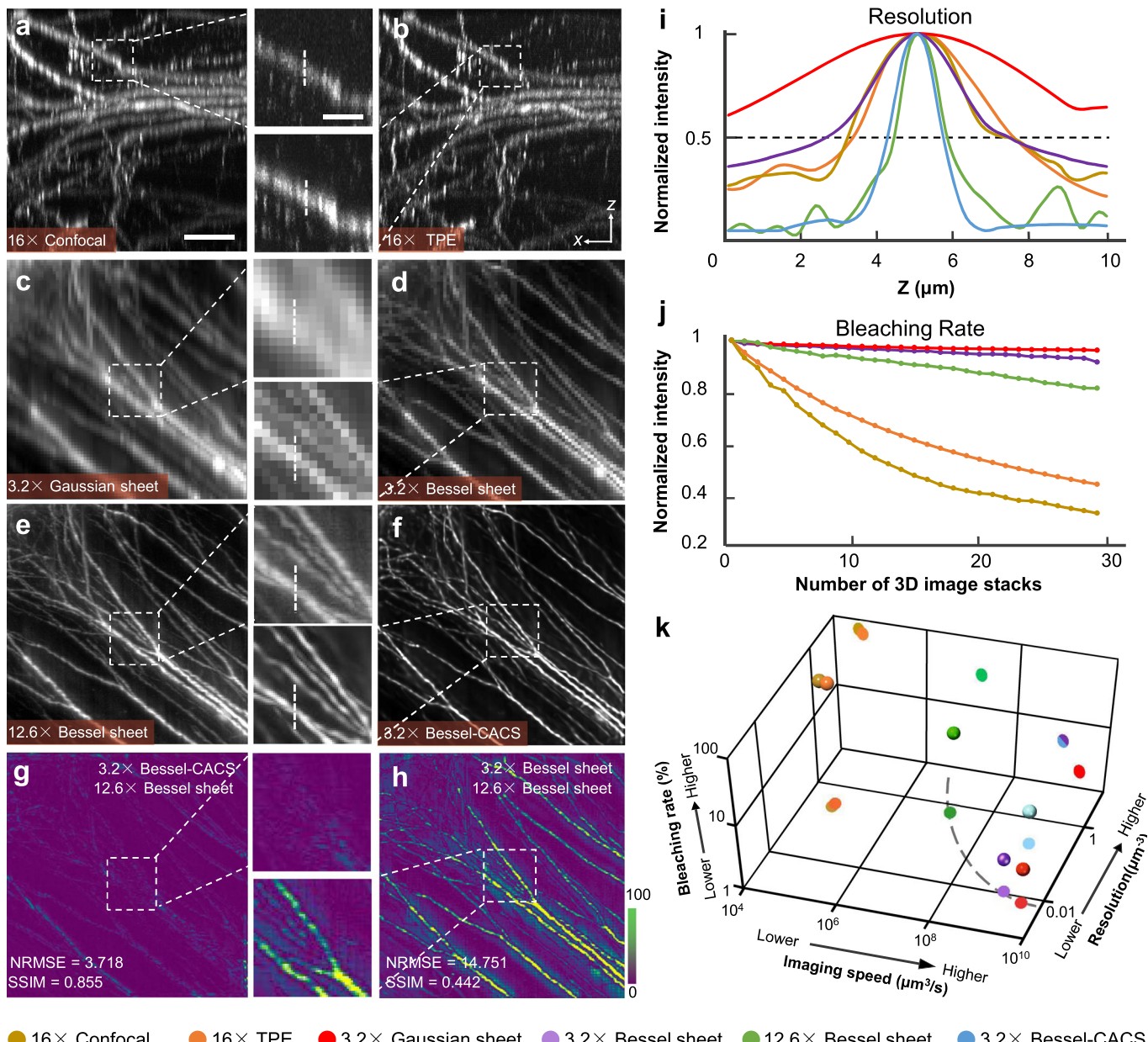

**Fig. 2 Comparisons of CACS Bessel sheet modes to confocal, TPE, and SPIM. a–f** Comparative image slices in an *x-z* plane through GFP-labelled neurons in a cleared mouse cortex by: point-scanning confocal microscopy (**a** ×16/0.8); two-photon excitation microscopy (**b** ×16/0.8); Gaussian sheet (**c** 0.02 NA illumination + ×3.2/0.28 detection); low-resolution Bessel light-sheet illumination (**d** 0.14 NA illumination + ×3.2/0.28 detection); high-resolution Bessel light-sheet illumination (**e** 0.28 NA illumination + ×12.6/0.5 detection); CACS Bessel sheet which successfully resolves the single neurons with using the same low-resolution setup (**f** 0.14 NA illumination + ×3.2/0.28 detection). Scale bar, 30 μm (inset, 10 μm). **g, h** Error maps of 3.2× Bessel-CACS output (**f**) and raw ×3.2 Bessel sheet result (**d**), as compared with ×12.6 Bessel sheet result (**e**). The calculated normalized root mean square error (NRMSE) and structural similarity (SSIM) further quantitatively confirm the high recovery fidelity of CACS as well as the resolution improvement. **i** Linecuts (as shown in insets in **a–f**) for each method, with 50% intensity level shown for estimation of the FWHM. **j** Bleaching rates obtained from repeated 3D imaging (30 times) of neurons in mouse cortex, normalized to account for the differences in SNR. **k** 3D plot comparing the acquisition speed, bleaching rate, and volumetric resolution of six imaging modes indicated by different colors. The dashed line that connects the data points of ×3.2 Bessel sheet, ×12.6 Gaussian sheet, and ×12.6 Bessel sheet projected in speed-resolution plane represents the throughput limit in conventional light-sheet microscopes. Source Data are available as a Source Data File.

yielded an isotropic resolution of ~1.5 μm at a high acquisition rate of ~1.5 mm³ s⁻¹. As shown in the data projected to a resolution-speed plane, the CACS Bessel sheet mode (light-blue point) breaks the SBP limit represented by the dashed resolution-speed curve, on which the data points of regular light-sheet modes have a relatively fixed speed/resolution value. Thus, it can be considered to be a tool facilitating the combination of large

FOV with isotropic high-resolution, which is an unmet need by other approaches.

**Rapid 3D imaging of single neurons in the whole brain at a timescale of minutes.** Our geometry with 1–5 μm thin tunable Bessel plane illumination and ×1.26–12.6 zoomable detection readily enables isotropic brain (or other organs) imaging at

different resolution scales (Supplementary Fig. 14). More importantly, the expanded SBP made possible by our CACS computation allows whole-brain (or other organs) imaging with isotropic subcellular resolution and at very high throughput, thereby quickly visualizing massive individual cells with details across the entire brain. In our demonstration of whole-brain imaging of neurons, a cleared mouse brain (Thy1-GFP-M, ~$10 \times 8 \times 5$ mm$^3$) was first imaged using the ×3.2 Bessel sheet mode (FOV 4.2 mm, Fig. 3a1) under two opposite views (0° and 180°), with totally 12 tiles (6 for each view) acquired to cover the entire brain (Fig. 3a2, Methods). Then, tile stitching[39] for each view (Fig. 3b1, 2) followed by a two-view fusion[40] was applied to obtain the 3D image of complete brain (Fig. 3a2, b3; 2-μm voxel). CACS was applied at the last step (Fig. 3a3) to generate a final output (0.5-μm iso-voxel size) with resolved nerve details comparable to those obtained with the ×12.6 Bessel sheet (Fig. 3b4, c, Supplementary Fig. 14, Movie 3). The 10-min rapid acquisition combined with parallel computation using multiple GPUs permits the efficient reconstruction of a digital brain with isotropic subcellular resolution over a 400 mm$^3$ volume (Fig. 3c, Methods, Supplementary Note Table 2, Movie 4). Magnified views of three small regions in cortex, cerebellum, and midbrain by ×3.2 Bessel sheet, ×12.6 Bessel sheet, and ×3.2 Bessel-CACS were also compared (Fig. 3c), revealing the various neuron morphology across the large-scale brain. The fine nerve fibers, such as apical dendrites, densely packed in these regions were successfully super-resolved in three dimensions by CACS procedure. Furthermore, through referring to the high-resolution ×12.6 Bessel sheet result, the recovery accuracy of CACS at whole-brain scale was also verified by comparing the NRMSE and SSIM values of 12 selected subregions ($100 \times 100 \times 100$ μm$^3$) across the entire reconstructed brain (Fig. 3d). Therefore, brain segmentations together with accurate quantifications could be implemented at system level (Fig. 3a4). For example, we precisely traced interregional neuron projections, which are important for understanding the functionality of the brain. Compared with the raw image, the CACS visualized more nerve fibers (Fig. 3c), thus enabling neuronal trajectories with abundant details to be presented in three dimensions (Fig. 3d, Supplementary Movie 3). After registering the CACS brain data with standard mouse brain atlas (Allen Brain Institute)[3,41] using a 3D image registration pipeline[42], the anatomical annotations were applied to the registered CACS brain image to create its 3D map (Supplementary Movie 5). The trajectories of five long-distance (LD) projection neurons were then semi-automatically traced in the map using Imaris, revealing how their pathways were broadcast across the anatomical regions (Methods, Supplementary Fig. 16). With isotropic submicron resolution, over 200 Pyramidal neurons densely packed at a cortex region could be also identified and traced automatically using established NeuroGPS-Tree method[43], as shown in Supplementary Fig. 15. Such quantitative analysis was implemented in a Thy1-GFP-M mouse with massive numbers of neurons being labelled, and this procedure would be even more efficient if the brain was more specifically labelled (e.g., by virus tracer).

**3D imaging and cell counting in half mouse brain**. In addition to the tracing of neuronal projections, the cyto-structures at different brain subregions were also explored using CACS Bessel sheet imaging (Fig. 4). In order to obtain numbers of cell in different subregions with the irregular and diversified morphologies of cells, we labelled nuclei with propidium iodide (PI), an organic small molecule staining DNA. Almost all cell nuclei including motor axons and retinal ganglion cells were stained, causing very dense fluorescing (Fig. 4a). Raw 3D images of the nuclei in half brain quickly obtained by 2× CACS sheet mode (Fig. 4a; in ~4 min), signals remained highly

overlapped and undistinguishable (Fig. 4b1). Identically, CACS offered a ×4 resolution enhancement in each dimension to recover images with resolution even better than ×8 Bessel sheet, to discern single nuclei (Fig. 4b2, Supplementary Movie 7). The vignette high-resolution views show that cell counting based on the CACS was as accurate as that using an ×8 Bessel sheet (Fig. 4c2, 3, Supplementary Movie 8). After anatomical annotation was applied by registering the image volume to the abovementioned Allen brain atlas (ABA), the labelled cells could be properly segmented (Fig. 4d–g, Supplementary Movie 9), with each individual cell being assigned an anatomical identity, such as isocortex, hippocampus formation, olfactory area, cerebral nuclei, cortical subplate, thalamus, hypothalamus, midbrain, hindbrain, or cerebellum, as shown in Fig. 4h, i. We did not observe morphology- or size-dependent errors in cell detection in different regions. By registering and annotating the detected cells to anatomical areas in the CACS Bessel Atlas, cell number and density information for these brain regions was quantified (Supplementary Figs. 17 and 18 and Movie 9). As shown in Fig. 4j, the total cell number in the half brain was ~$3.3 \times 10^7$, and the average density was ~$2.4 \times 10^5$ cells per mm$^3$. Among the primary brain regions, the Isocortex showed the highest number of cells at $8.2 \times 10^6$, which compares with $2.7 \times 10^6$, $8.0 \times 10^6$, $4.7 \times 10^5$, $3.3 \times 10^6$, $2.4 \times 10^6$, $1.2 \times 10^6$, $2.5 \times 10^6$, $1.3 \times 10^6$, and $3.3 \times 10^6$ cells in the Olfactory area (OLF), Cerebellum (CB), Cortical subplate (CTXsp), Cerebral nuclei (CNU), Hippocampal formation (HPF), Hypothalamus (HY), Midbrain (MB), Thalamus (TH), and Hindbrain (HB), respectively. The cerebellum had the highest cell density of ~$4 \times 10^5$ cells per mm$^3$, compared with $2.5 \times 10^5$, $2.4 \times 10^5$, $2.3 \times 10^5$, $2.1 \times 10^5$, $2.2 \times 10^5$, $2.6 \times 10^5$, $2.2 \times 10^5$, $2.0 \times 10^5$, and $1.7 \times 10^5$ in the other regions listed above. Although the current results were obtained using a PI-stained half brain, in which ~50% of all motor axons and retinal ganglion cells were counted, our results are consistent with previously reported work[2].

**Dual-color 3D imaging and quantitative analyses of neuromuscular junctions**. We further demonstrated dual-color CACS Bessel sheet imaging of motor endplate (MEP, α-BTX) and peripheral nerves (Thy1-YFP) in mouse muscles. The overall spatial distribution of MEPs in the muscle tissue and the detailed neuromuscular junction (NMJ) occupancy defined as the volume ratio of the underlying postsynaptic MEPs (red) occupied by presynaptic vesicles (green), were both visualized and analyzed using the CACS Bessel sheet microscopy. We first finished high-throughput screening (×3.2 Bessel sheet) of two gastrocnemius muscles and two tibialis anterior muscles (~ 720 mm$^3$ in total) rapidly in ~20 min (Fig. 5a–d, Supplementary Movie 10). The image-based segmentation of the MEPs and nerves across whole-muscle scale was then enabled as a result of the quick 3D visualization (insets in a–d, Supplementary Movie 11). Image results of two ROIs in gastrocnemius (Fig. 5a) by raw ×3.2 Bessel sheet, ×3.2 Bessel-CACS, and high-resolution ×12.6 Bessel sheet are compared in Fig. 5e, to show the quality of ×3.2 Bessel-CACS results being as high as those from ×12.6 results. The magnified views of single NMJs (six white boxes) further show that CACS could reveal fine neuromuscular structures in single NMJs (Supplementary Movie 11). The number of MEPs and their density in each muscle were thereby quantified based on the large-scale 3D images (Fig. 5f). We found that although the total MEP number varied in each muscle, their density remained relatively close. Furthermore, owing to the high spatial resolution achieved across whole-muscle scale, our method also permitted the accurate measurement of presynaptic and postsynaptic structural volumes in single NMJs and the consequent calculation of NMJ occupancy (right, Fig. 5g), which are not possible for conventional ×3.2 Bessel results (left, Fig. 5g). These complete visualization and quantitative analyses enabled by CACS Bessel sheet imaging indicate both the contraction

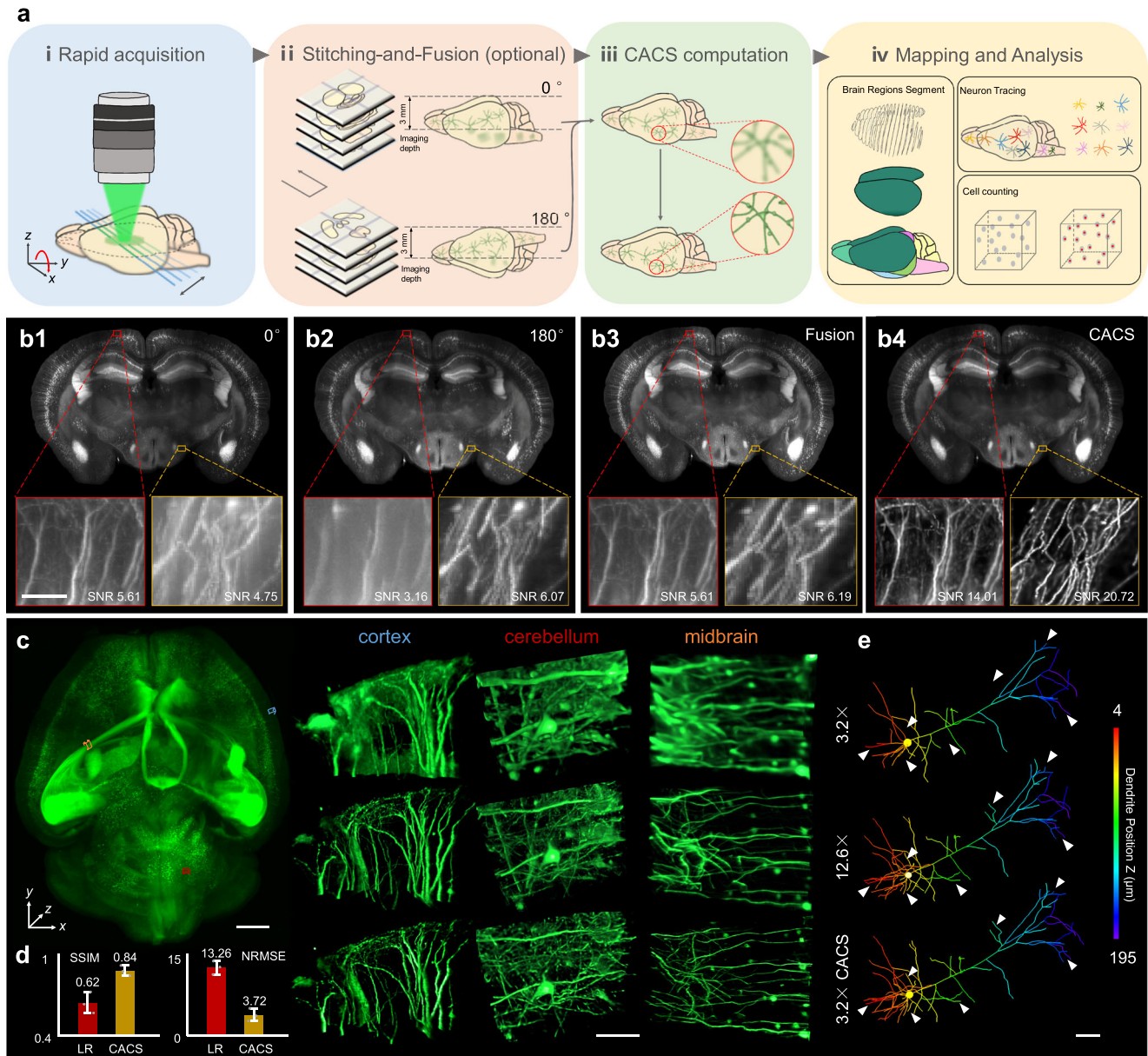

**Fig. 3 Whole-brain mapping pipeline. a** Flow chart of whole-brain data acquisition and processing, which includes: (i) rapid image acquisition by ×3.2 Bessel sheet, with total 12 tiles under 2 views acquired in ~10 min. (ii) tile stitching for each view followed by 2-view weighted fusion, to reconstruct a scattering-reduced whole brain at low resolution (2-µm voxel). The full implementation of this step would be not necessary if the sample is optimally cleared or imaged under lower magnification. (iii) CACS computation to recover a digital whole brain with improved resolution (0.5-µm iso-voxel size) and SNR. (iv) quantitative analyses, such as brain region segmentation, neuron tracing, and cell counting, based on high-quality whole-brain reconstruction. **b** MIPs of reconstructed coronal plane (x-z, 150-µm thickness) in whole brain by 0°, 180° single view Bessel sheet, 2-view fusion and CACS computation. The magnified views of two small regions selected from $z = 100$ and $z = 4500$ µm depth further show the effect of 2-view fusion and CACS. Scale bar, 50 µm. **c** 3D visualization of a whole digital mouse brain (left, ~400 mm³, 3-trillion voxels) reconstructed from the ×3.2 Bessel-CACS results, whose raw data were rapidly acquired in ~10 min. Three magnified volumes from the cortex (blue), cerebellum (red) and midbrain (orange) regions by raw ×3.2 Bessel sheet (top row), ×3.2 Bessel-CACS (middle row), and ×12.6 Bessel sheet (bottom row) modes are compared to show the super-resolution capability of CACS procedure. Scale bars, 1 mm for whole-brain visualization, 50 µm for ROIs. **d** SSIM and NRMSE values of 12 brain subregions (100 × 100 × 100 µm³) images by raw Bessel sheet and Bessel-CACS. The quantitative analyses verify a high recovery fidelity of CACS results as compared to the high-resolution ground truths. Data are presented as mean values ± SD. **e** Tracing of a single pyramidal neuron under three modes. The same neuron imaged by three modes was segmented and traced using Imaris. It's shown that CACS result enables the identification of more abundant neuron substructures. Scale bar, 50 µm. Source Data are available as a Source Data File.

capability of muscle tissues at a macroscale, and the delivery efficiency of neurotransmitters at nerve endings on a micro-scale, thereby potentially benefiting the comprehensive understanding of neuromuscular functional activities and improving the diagnosis and development of therapy for related diseases[44,45].

## Discussion

In summary, the marriage of the CACS strategy with long-range Bessel plane illumination enables quick 3D imaging of millimeter-to-centimeter sized samples in seconds to minutes, with sub-cellular isotropic resolution and high throughput of up to 10

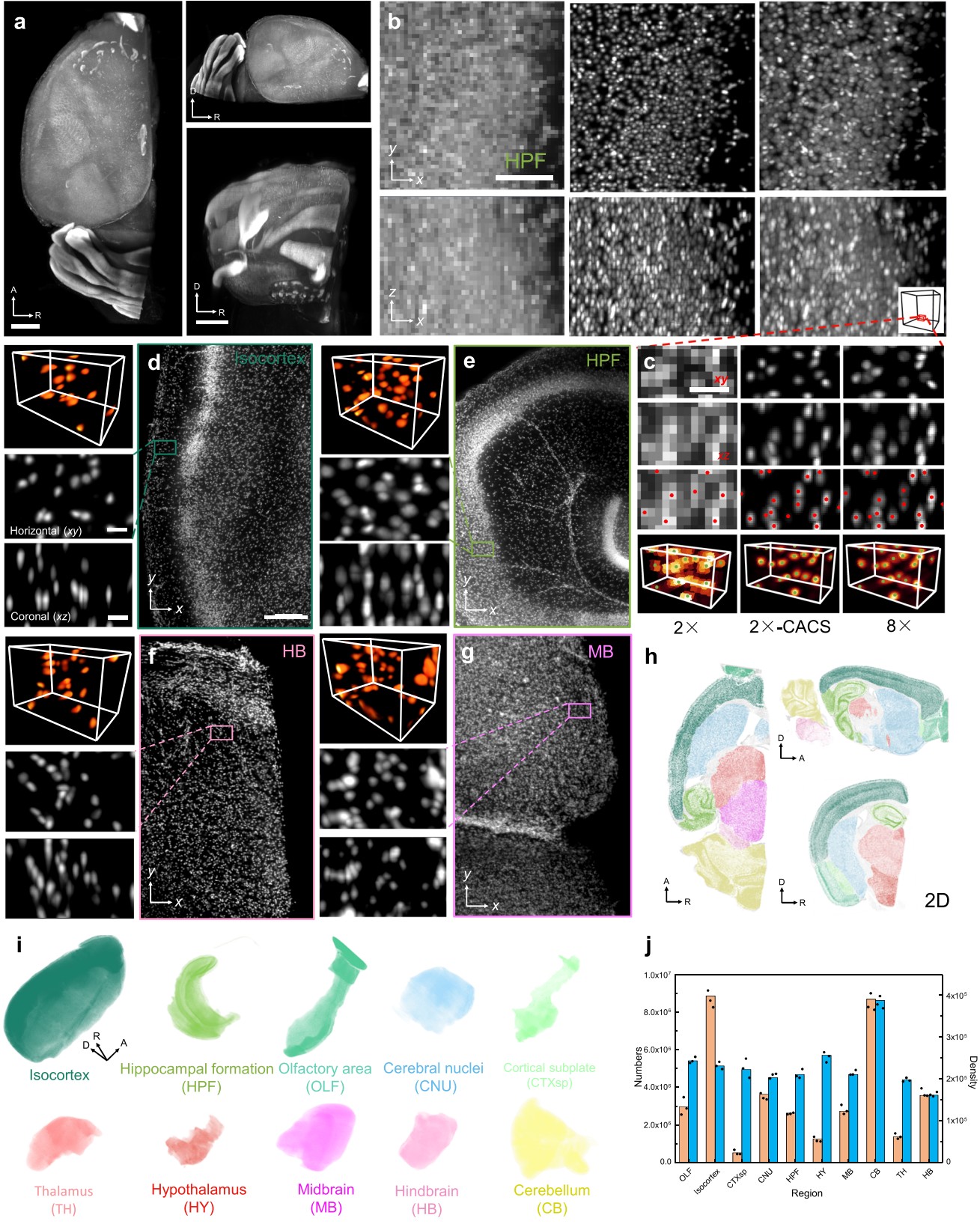

gigavoxels per second. As the size of optically cleared and labelled sample becomes larger and larger nowadays, such technical advances of resolution/throughput shown in our method also becomes increasingly valuable, especially for the *intoto* mapping of entire organs/organisms at variable scales. Highly inclined LSM

has been used for the 3D anatomical mapping of thick organs where autofluorescence and out-of-focus excitation would otherwise be prohibitive under wide-field illumination. With the thinner light-sheets of the Bessel beam, the effective FOV would be less compromised, and axial resolution would be further

**Fig. 4 Cell counting of propidium iodide (PI)-labelled half mouse brain. a** Coronal view, transverse view and sagittal view of PI-labelled half mouse brain image by ×2 CACS Bessel sheet. Scale bar, 1 mm. **b** A selected ROI in HPF, resolved by ×2 Bessel sheet (left column), ×2 CACS Bessel sheet (middle column), and ×8 Bessel sheet (right column). **c** Magnified views of a small volume (red box) in **b**, showing the different accuracy of nuclei based identification base on three types of images. Scale bar, 20 μm in **b** and 10 μm in **c**. **d–g** 3D visualization of four high-resolution ROIs from Isocortex, HPF, HB and MB regions in the reconstructed half brain. Scale bar, 50 μm in large box and 10 μm in small box. The results clearly show single-cell resolution in three dimensions, which is sufficient for cell counting. **h** 10 manually segmented brain regions in the half brain, shown in transverse, coronal, and sagittal views. **i** 3D visualization of the 10 subregions indicated by different colors. **j** Calculated nuclei number and density of each subregion in half brain. Data are presented as mean values ± SD (*n* = 3 biologically independent samples). Source Data are available as a Source Data File.

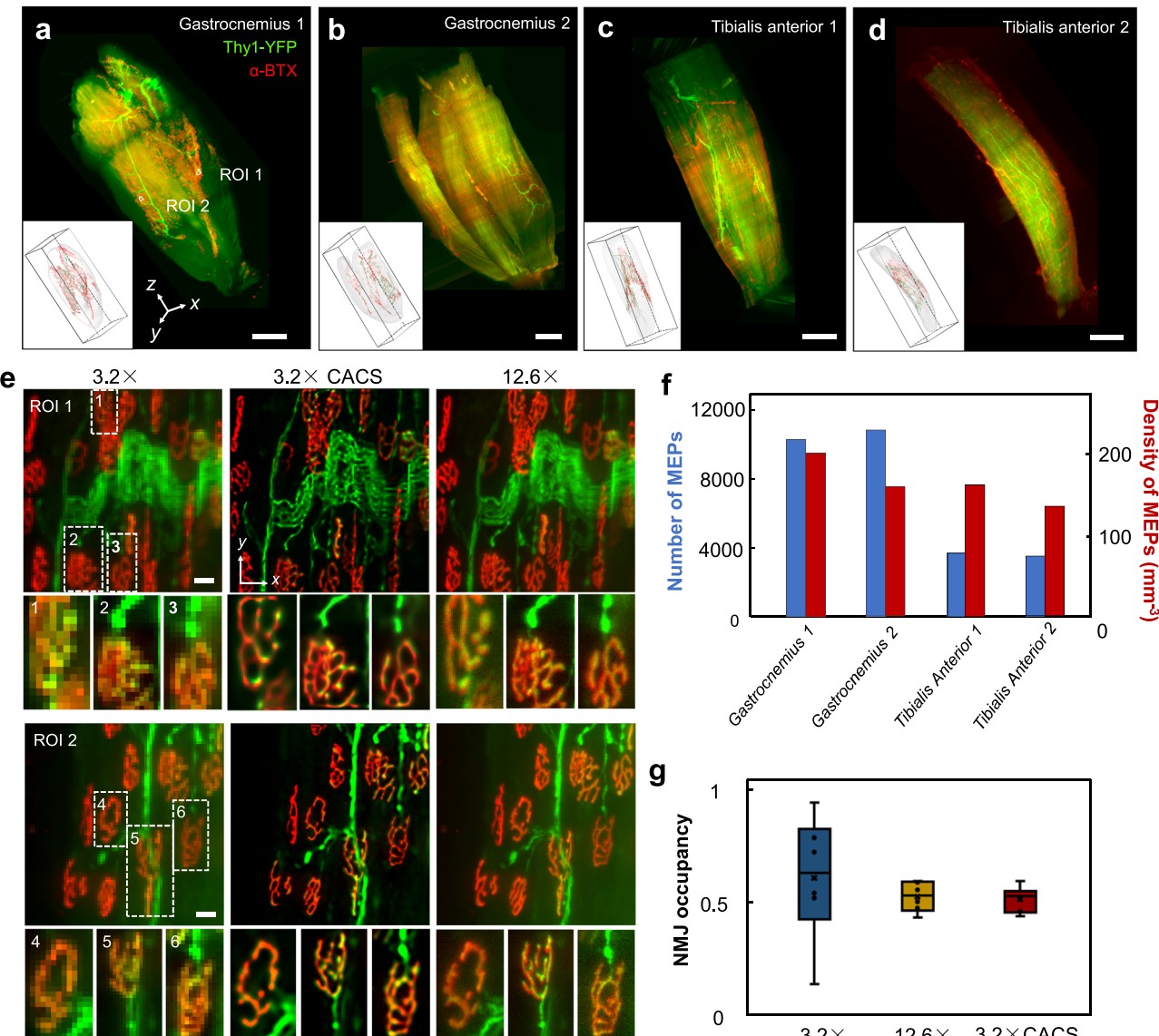

**Fig. 5 Dual-color CACS imaging and quantitative analyses of neuromuscular junctions in mouse Gastrocnemius and Tibialis anterior. a–d** Dual-color, 3D visualization of Gastrocnemius and Tibialis anteriors, showing both the presynaptic terminal bouton (green, Thy1-YFP) and postsynaptic acetylcholine receptors (red, α-BTX). Insets show image-based segmentation of presynaptic terminal and MEPs. **e,** Comparison of two ROIs (100 × 100 × 100 μm³) in a, by ×3.2 (left column), ×3.2 CACS (middle column), and ×12.6 Bessel sheet (right column). Six magnified views (white boxes, 1, 2, 3 for ROI1 and 4, 5, 6 for ROI2) further demonstrate the structural information of single NMJs. Scale bar, 10 μm. **f** Macroscale quantitative analysis of MEP, including total numbers and densities of MEPs, in two types of muscles (*n* = 4 biologically independent samples). **g** Neuromuscular junction (NMJ) occupancy calculated based on abovementioned ×3.2, ×3.2 CACS, and ×12.6 Bessel sheet images that show different level of resolution. (maxima/minima: the maximum/minimum values; centre: the medium value; bounds: upper and lower quartile) The result from raw ×3.2 image shows an unacceptably large error range, owing to the ambiguous cellular resolution. Here NMJ occupancy was calculated as (presynaptic volume/postsynaptic volume) ×100%. Data are presented as mean values ± SD (*n* = 6 biologically independent samples). Source Data are available as a Source Data File.

improved. The introduction of dual-side illumination and dual-view detection would further reduce light depletion in thick tissue and permit the complete interrogation of entire organs. As currently configured, the key step to maintaining thin and wide Bessel plane illumination under a large FOV is to synchronize the scanned beam with the confocal electronic slit of the camera, to eliminate excitation from the side lobes. This could possibly be improved by direct generation of a side-lobe-reduced Bessel sheet using recent masking techniques[46]. At the other extreme, the possible introduction of two-photon excitation may be furthermore suited for improved depth penetration when imaging more scattering samples.

Despite the use of plane illumination, current microscopes have throughput limited to megavoxels per second. This causes issues in 3D anatomical mapping, in which trillion-voxel data are frequently required. Therefore, the long-time acquisition required for mechanical image stitching prevents a variety of histological, pathological, and neuroanatomical research being implemented on a large scale. Our CACS procedure uses a single image stack to three-dimensionally improve the contrast and resolution limited by the pixilation and numerical aperture, computationally transforming our Bessel microscope into a gigavoxel-throughput-boosted imager that can achieve isotropic 3D super-resolution imaging of mesoscale samples without the pain of a long acquisition lasting for hours to days. It should be noted that though CS processing is efficient for the recovery of under-sampled signals, a condition satisfied in our low-magnification large-FOV imaging setup, it remains challenging for regular CS implementations to handle the large-scale image appropriately because of the uncertainty in the regularization applied to the highly varying signals. Our introduction of a content-aware feedback strategy can reasonably extract the signal characteristics and thereby calculate the optimal regularization parameter for each image block automatically, allowing recovery of signals on a large scale with minimal artefacts as well as obvious resolution improvement (Supplementary Note Fig. 1). Therefore, while CS is known to have certain limitations for processing signals containing complicated structural information[47], its effects on line-like neurons and point-like nuclei were notably improved after the content-aware adaption to the signals involved. Our 3D CACS computation is unsupervised, without the need for prior training process with a higher-resolution database, and highly parallelized for GPU acceleration. Furthermore, it's also applicable to other classic 3D imaging approaches, such as confocal and TPEM, with increasing their optical throughput as well (Supplementary Fig. 11). We expect these performance advantages could broadly benefit the implementation of high-throughput organ/organism mapping for neuroscience, histology, pathology studies, and potentially bring new insights for computational imaging techniques, so that we can keep pushing the optical throughput limit to extract ever more spatiotemporal information from biological specimens.

## Methods

**Zoomable Bessel plane illumination microscopy setup**. The optical setups of zoomable Bessel plane illumination microscope were detailed in Supplementary Figs. 1 and 2, Movie 1. Our Bessel plane illumination microscope (Supplementary Fig. 1, Movie 1) sweeps a long Bessel beam in the $y$ direction across the focal plane of the detection objective to create a large-scale scanned light-sheet and yield an image at a single $z$ plane within the specimen. To maintain sufficient laser intensity at large scale, we used a large apex axicon (176e, AX252-A, Thorlabs) instead of mask to create annular beam at a plane conjugate to both the galvanometer and the rear pupil of the excitation objective. For the Bessel beam, its axial extent of central maximum (beam thickness) and diffraction distance (beam length) are proportional to the magnification factor of the illumination objective and its square, respectively. To suppress the laser attenuation/scattering from samples, we introduced two opposite plane illumination sources from dual sides of the brain. A macro-view microscope (Olympus MVX10, ×0.63 to ×6.3, combined with a ×2 detection objective lens MV PLAPO

2XC) providing zoomable magnification from ×1.26/0.14 to ×12.6/0.5, together with an sCMOS camera (Hamamatsu ORCA-Flash4.0 V2) were used as the fluorescence detection unit, providing a tunable FOV from 1 to 10 mm and effective lateral resolution from ~1 to 10 μm for large specimens. To achieve isotropic resolution, we used a set of switchable long working distance objectives (Mitutoyo, ×2/0.055, or ×5/0.14, or ×10/0.28) from each side to form a tunable dual-side illuminating Bessel beam with axial extent and diffraction length matched with the varying lateral resolutions and FOV, respectively. To achieve side-lobe free imaging based on electronic confocal slit, we formed a linear array, which contains 2–4 rows of active pixels sweeping at the sensor plane along $y$ direction, and tightly synchronized with the central maximum of scanned Bessel beam (Supplementary Movie 1). During image acquisition, the 2D galvo mirror (GVS012, Thorlabs) that scanned the Bessel beam into plane, the sCMOS camera that continuously recorded the fluorescence images at high speed up to ~50 frames per second, the motorized $xyz$ stage (Z825B, Thorlabs combined with SST59D3306, Huatian) that moved the sample in three dimensions, and the motorized flip mirror that switched the illumination between dual-side beams, were all controlled by analog signals generated by a data acquisition device (PCIe-6259, National Instruments). To enable high-throughput volumetric imaging of specimen, a customized sample holder rapidly moved the sample across the light-sheet along $z$ direction, with plane images at different depths consecutively obtained (Supplementary Fig. 2). After imaging under the first view (0°), the sample was flipped using a micromotor integrated into the holder (Supplementary Fig. 2), and imaged under the second up-side-down view (180°). It's noted that though the current system was optimized for the amber-like harden samples made by dehydration-based clearing methods (UDISCO, PEGASOS, etc.), it can also handle other types of cleared samples, e.g., CUBIC brain, via a retrofit of holder design. A LabVIEW (National Instruments) program was developed to synchronize the motorized parts, to automatically realize the line-synchronized scanning, $z$-scanning, tile stitching, and sample rotation in order.

**CACS computation procedure**. If the original signals are sparsely distributed in the image volumes or can be sparsely represented by some given appropriate basis (e.g., in Fourier, or Wavelet space), CS allows the recovery of them from incomplete measurement[32], which is our case in large-FOV Bessel sheet imaging. Since the intensity of signal in mouse brain has a large dynamic range (Supplementary Figs. S6–9), it can be sparsely distributed and recovered in the Fourier domain effectively. Regarding Eq. 1 running at Fourier space, the term $Ax_i - y_i$, which represents the difference between the low-resolution measurement $y_i$ and the high-resolution output $x_i$ degraded by the measurement matrix $A$, needs to be calculated during each iteration. Thus, an accurate $A$ is required here and can be obtained by the Fourier transformation of the system PSF. Then each $x_i$ can be recovered from the low-resolution $y_i$ through iteratively solving the Eq. 1 using an interior-point method[48] (Supplementary Note 5):

$$\text{minimize} \|Ax_i - y_i\|_2^2 + \lambda_i \|x_i\|_1 \qquad (1)$$

The content-aware weighting factor of regularization $\lambda_i$ is introduced to make the solution of equation adaptive to the signal characteristics. $\lambda_i$ dynamically balances the image results from more like the original measurement to more like the CS recovery. The value of $\lambda_i$ is calculated by the multiply of $\alpha_i$ and $\beta_i$, where $\alpha_i$ is an index indicating the signal density of input image (Supplementary Note 3), and $\beta_i$ is inversely proportional to the entropy, indicating the amount of information in input image (Supplementary Note 3). Therefore, the choice of $\lambda_i$ is highly adapted to the signals, which could vary a lot at large scale. Our content-aware processing procedure divides the raw image, e.g., half brain image $Y$ with dimension $10 \times 4 \times 5$ mm, into a series of small volumes, e.g., blocks with $100 \times 100 \times 100$ voxels in which the signals can be considered as uniformly distributed, and then calculates the regularization factor for each volume to obtain its optimal CACS result $x_i$. After the iterative recovery, multiple obtained $x_i$ are transformed back to high-resolution image blocks $X_i$ in spatial domain, which would be further stitched into the final output $X$ (Supplementary Movie 2). To boost the processing speed, CACS computation can be further parallelized using multiple-GPU acceleration.

**Rapid 3D isotropic imaging of mouse brain at various scales**. The improved axial resolution obtained from the tunable Bessel plane illumination and zoomable detection enables easy light (or other organs) imaging at isotropic resolution and various scales. As a reference, four clarified mouse brains (Tg: Thy1-GFP-M) were quickly screened at a throughput of one brain min$^{-1}$ and ~10 μm isotropic resolution (Supplementary Fig. 14b). Each brain ($10 \times 8 \times 5$ mm) was imaged under the lowest ×1.26 with our MVX10 microscope. Two ×2/0.055 excitation objectives generated a Bessel light-sheet with a thickness of 5 μm from dual-side. Considering ultrahigh resolution in fast screening mode was not required, we set the scanning step size as 5 μm (voxel size $5.16 \times 5.16 \times 5$ μm), to final achieve ~10 μm isotropic resolution. Under the ×1.26 dual-side Bessel sheet mode, we acquired a 3D image stack containing ~600 whole-brain slices at each side in each view, finally obtaining four image stacks containing over 2000 large-FOV image slices for both sides of both views in ~60 sec. These four sets of image stacks were then computationally stitched and fused together, forming a final 3D image of the whole brain in a few

minutes[2]. By obtaining the coarse 3D structures of the whole brains using the widest Bessel sheet (Supplementary Fig. 14b), the transverse (x-y), coronal (x-z), and sagittal (y-z) views could be extracted from the reconstructed brains to quickly identify brain regions where desired signals were present (Supplementary Fig. 14c). Then, higher-resolution imaging of any region of interest was possible using the ×12.6 Bessel sheet mode. For example, No. 2 brain with showing the best signal distributions was chosen after the fast screening. We then imaged three ~$8 \times 10^{-3}$ mm³ regions of interest (ROIs) in the cortex, hippocampus, and cerebellum of No. 2 brain (Supplementary Fig. 14b) at an imaging speed of ~0.01 mm³ s⁻¹ and an isotropic resolution of ~1.5 μm, respectively. For ROIs imaging under ×12.6 magnification, we switched the motorized flip mirror (MFF101/M, Thorlabs) to choose single side illumination (left or right), depending on the position of ROIs. The scanning step size was 0.5 μm for sufficient z sampling. Therefore, the voxel size was $0.516 \times 0.516 \times 0.5$ μm. Vignette high-resolution views of these volumes are shown in Supplementary Fig. 14d–f, detailing the various neuron morphologies across the brain. Besides the identification of different types of neuron cell bodies (e.g., pyramid neurons in Supplementary Fig. 14e, and astrocytes in Supplementary Fig. 14f), nerve fibers such as densely packed apical dendrites (Supplementary Fig. 14d) were also clearly resolved in three dimensions.

**Whole-brain CACS imaging procedure.** For high-resolution mapping of whole mouse brain to enable single-neuron-level quantitative analysis across whole brain, the choice of repetitive tile stitching, e.g., around 100 times under ×12.6 magnification, is not only very time consuming, estimated with over 15 h, but also brings substantial photobleaching. Instead, we rapidly imaged the whole brain with lower ×3.2 Bessel sheet mode, and used CACS to computationally improve the compromised resolution. With a $4.16 \times 4.16$ mm imaging FOV, we sequentially obtained six tiles (3 rows × 2 columns) to cover the entire brain. During image acquisition, the dual-side laser-sheets illuminate the sample not in a simultaneous but a sequential fashion (left laser for three tiles in left column, and then right laser for another three tiles in right column), to increase the laser power and minimize the light scattering from the far side illumination. The imaging depth for all the tiles is 0–3 mm (1500 frames, 2 μm step size). The tissues deeper than 3 mm (up to ~5 mm) were not imaged, to avoid the acquisition of scattered signals and also reduce the acquisition time. The six tiles were stitched together to generate a large-scale 3D image with size of $11.4 \times 7.8 \times 3$ mm³. As previously mentioned, such tile imaging was implemented twice under 0° and opposite 180° views. Then the stitched dual-view images with ~$11.4 \times 7.8 \times 1$ mm³ overlap were registered and fused using a Fiji registration plugin[40]. During this process, the 3D transformation matrix between the two views was determined by registering the signals in the overlapped region. An affine transformation model including rotation, translation and deformation allows precise iterative registration of the 0° and 180°-view images. Finally, a whole-brain image with size of $11.4 \times 7.8 \times 5$ mm³ was created through a weighted fusion of the two registered views. The whole-brain data encompassed complete structural information with achieving ~4-μm spatial resolution (2-μm voxel), which remained insufficient for tracing of fine neuron projections, or counting of densely-labeled cell nuclei. Nevertheless, CACS was then applied with automatically subdividing the whole-brain data into thousands of blocks for being processed using 4 GPUs (RTX2080ti) in parallel block by block ($100 \times 100 \times 100$ voxels for each block), owing to the memory limit of the GPUs. Finally, the super-resolved patches were automatically stitched together into a complete 3.5 trillion-voxel image (0.5-μm iso-voxel size) with an acquisition time ~10 min, which was compared to tens of hours by other methods[1], and a computation time ~5 h, which was estimated similar with the time consumption for stitching-and-fusion under ×12.6. A 16T SSD RAID0 (SAMSUNG 2T × 8) was created to guarantee the high-speed implementations of image stitching, registration, fusion, and CACS computation without much I/O delay. It should be noted that when visualizing the brain data using Imaris, the *ims* file realized hierarchical resolution when zooming in-and-out the image at different magnifications. An ordinary cellular resolution was enough for viewing the overall appearance of whole brain, while a subcellular resolution was provided when observing neuronal details in a small ROI. Such hierarchical-resolution display could greatly save the data storage and I/O time. For example, the ims file of a high-resolution whole brain occupied less than 2 TB space, which was compared to original 6 TB for tif data.

It should be noted that the combination of Bessel sheet with CACS is very flexible, providing throughput improvement for zoomable imaging from ×1.26 to ×12.6. It is also easy to modify the LabVIEW program to conduct imaging under different modes/magnifications. Thus, it's beyond the imaging of brain we demonstrate, being widely suited for anatomical imaging of various large samples, such as mouse muscles shown in Fig. 5.

**Dual-channel mouse muscle tissues imaging procedure.** For dual-color imaging of gastrocnemius muscles and tibialis anterior muscles, we excited neurons and presynaptic membranes with a 488 nm laser and postsynaptic membranes (also known as MEPs) with a 637 nm laser (RGB-637/532/488-60 mW, Changchun New Industries Optoelectronics Technology Co., Ltd.). Under ×3.2 Bessel sheet mode, we stitched twice and three times for entire tibialis anterior muscle and gastrocnemius muscle, respectively. The thickness of all the muscles were around 3 mm, hence the imaging under two views for multiview-fusion was not necessary. It

merely took around 5 min to complete dual-channel imaging of each muscle, and the total acquisition time was approximately 20 min.

**Sample preparation.** UDISCO[19] clearing method was used to clarify the dissected wild-type mouse brain with cell nuclei stained by PI dye, neuron-labelled mouse brain blocks (Tg: Thy1-GFP-M). FDISCO clearing method[49] was used to clarify the neuron-labelled mouse muscles (Tg: Thy1-YFP-16) with α-BTX tagging the motor endplate. PEGASOS clearing method[50] was used to clear the Thy1-GFP-M whole mouse brain, The cleared organ samples were harden and thus could be firmly mounted onto our customized holders (Supplementary Fig. 2). The BABB-D4 for UDISCO, DBE for FDISCO or BB-PEG for PEGASOS immersion liquid was filled in the sample chamber for refractive index-matched imaging with least aberration. For measuring the system's point spread function, fluorescent beads (Lumisphere, 1%w/v, 0.5 μm, Polystyrene) were embedded into a specifically formulated resin (DER332, DER736, IPDA, 12: 3.8: 2.7), the refractive index of which was equal to BABB-D4, to form a rigid sample that can be clamped by the holder and imaged in the chamber.

**Whole-brain visualization and brain region registration to ABA.** After the CACS enhancement on the low-resolution whole-brain data, we used an adaptive registration method to three-dimensionally map the brain to the standard Allen Brain Atlas (ABA). Since ABA was reconstituted from a series of coronal slices, we also re-orientated our recovered image from horizontal view to coronal view and thus prealigned it to the ABA. This prealigned brain was first downsampled into a low-resolution version (20-μm voxel), which was quickly registered with the ABA using an Elastix-based bichannel image registration pipeline[42]. The obtained transformation correspondence was then applied to the Allen anatomical annotations to generate the transformed annotations of our Bessel brain, with which a Bessel brain map could be created. A manual adjustment was followed to further improve the registration/annotation accuracy. Meanwhile, Bessel brain image with native resolution was visualized in Imaris to facilitate the neuron and cell number analyses. Neuron tracing and cell counting were operated at high resolution and without the need of being registered to ABA. The segmented neurons/cells nuclei were then merged with the 3D brain map that was readily created at low resolution. Their trajectories/localization at encephalic regions (such as isocortex, hippocampus, cerebellum and midbrain) could be *intoto* mapped out at a whole-brain scale (Fig. 4h, i, Supplementary Movie 7 and 9). The pathways of several long projection neurons across various brain subregions were identified and annotated, as shown in Supplementary Fig. 16. The neuron population/density in different encephalic regions were also quantified by calculating the volume of the regions and counting the identified cell bodies within them (Fig. 4j, Supplementary Movie 9).

**Neuron tracing.** The neurons in whole-brain data were segmented semi-automatically using the commercial Imaris software. With registering our brain to ABA, we obtained the anatomical annotation for all the segmented areas. The Autopath Mode of the Filament module was applied to trace long-distance neurons. We first assigned one point on a long-distance neuron to initiate the tracing. Then, Imaris automatically calculated the pathway in accordance with image data, reconstructed the 3D morphology, and linked it with the previous part. This procedure would repeat several times until the whole neuron, which could also be recognized by human's eye, was segmented. Furthermore, we also tried automatic tracing of massive neurons at a cortex region using Neuron GPS-tree approach[42]. The tracing procedure mainly included the soma detection and neuron tracing. The raw image volume was first imported into a pre-processing program to detect the somata, which were the sources for neuron tracing. The exported file indicating the location and size information of somata were then input to the main tracing program. At tracing step, we again imported the raw image volume and repeated the soma detection operation to also identify the neurons fibers. Then an intensity threshold was set to determine the endpoint for the neuron tracing. The final traced output was saved as swc format file, which could be loaded into Amira software for visualization or being further analyzed.

**Cell counting.** The Spots module and Surface module of commercial Imaris software was used to count cells in various anatomical regions of CACS-reconstructed half brain (full resolution, ~400 gigavoxels). We first separated several primary brain regions into different channels in Surface module. Then automatic creation in Spots module was applied to count cells number for each single channel which represents an encephalic region. To achieve accurate counting, the essential parts were the appropriate estimate of cell bodies' diameter and filtration of the chosen cells by tuning the quality parameters. The accuracy of this automatic counting procedure was also verified by researcher's visual inspection, which herein severed as ground truth. The averaged error rate was <5% for high-resolution CACS results (n = 20 volumes). After obtaining the total number of cells in each primary brain region, according to the subregion ranges divided by the Surface module, Imaris could also calculate the volume of each primary brain region. Then, with knowing the number of cell nuclei and the volume of each segmented brain region, the density of the cell nuclei inside each brain region could be calculated.

**Counting of MEPs and calculation of NMJ occupancy.** The method to count MEPs was similar to the abovementioned cell nuclei numbers counting with PI-labelled mouse half brain. Under the low-resolution ×3.2 magnification, the optical

blur brought by the low NA and low sampling rate caused the details inside a single MEP to be lost, and the entire MEP blurred to be a point, which in turn helped us use Imaris' Spot module to count the number. Considering that we only needed the total number and overall density inside the entire muscle, segmentation was no longer necessary. Select the entire region contained in the image and set the threshold according to the diameter of the single MEP, then we could get the total number of MEPs within a single muscle tissue. To calculate the overall density, we needed to calculate the volume of the whole muscle by the method mentioned before.

Calculating NMJ occupancy required accurate measurement of presynaptic and postsynaptic structural volumes in a single NMJ. After importing the dual-channel images (YFP and α-BTX) into Imaris, we randomly selected 10 NMJs for analysis. For each NMJ, we used the Surface module to calculate the volume of the presynaptic and postsynaptic structural volumes acquired by different channels. The two volume results were divided to obtain the occupancy of a single NMJ.

**Reporting summary**. Further information on research design is available in the Nature Research Reporting Summary linked to this article.

## Data availability
The datasets generated and analyzed in this study are available in the https://doi.org/10.6084/m9.figshare.13220864 and Supplementary Information, and also are available from the corresponding authors upon request. Source data are provided with this paper.

## Code availability
Custom codes for CACS computation implemented in current study are available at https://github.com/fangchunyu/CACS-Demo (https://doi.org/10.5281/zenodo.4263542).

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

## Acknowledgements

We thank Shangbang Gao, Luoying Zhang, Haohong Li, Man Jiang, Bo Xiong for discussions and comments on the manuscript, Yunyun Han and Yao Zhou for the discussion on the imaging and brain visualization, Zhilong Yu and Hao Zhang for the support on the code implementation. This work was supported by the National Key R&D program of China (2017YFA0700501, D.Z. and P.F.), the National Natural Science Foundation of China (21927802, 21874052 for P.F., 61860206009 for D.Z., 81873793 for W.M.), the Innovation Fund of WNLO (P.F. and D.Z.), and the Junior Thousand Talents Program of China (P.F.).

## Author contributions

P.F. conceived the idea. P.F., D.Z., and W.M. oversaw the project. C.F. and X.W. developed the optical setups. C.F. developed the programs. C.F., T.C., T.Y., Y.H., W.F., F.Z., P.W., and Y.L. conducted the experiments, processed the images, and visualized the data. C.F., T.Y., D.Z., W.M., and P.F. analyzed the data and wrote the paper.

## Competing interests

The authors declare no competing interests.
