## [Peer Review File · Nature Communications]

Reviewers' Comments:

Reviewer #1:

Remarks to the Author:

Described is a wide field of view light sheet microscope that uses a scanned Bessel beam to selectively illuminate a thin plane in samples such as whole mouse organs. Unlike Gaussian light sheets, the Bessel beam (which is long and thin) does not require axially sweeping of the waist in order to obtain a sharp image throughout a large field of view. This allows for a substantial increase in acquisition speed. Furthermore, Bessel beams have "self healing" properties as they propagate: this makes them more immune to scattering than a Gaussian beam and so should improve image contrast in thick samples.

The authors augment their microscope using a computational pipeline: content-aware compressed sensing (CACS), which up-scales image resolution. CACS seems like a substantial improvement over conventional deconvolution techniques. A whole brain imaged at a pixel size of $2 \times 2 \times 2$ microns was upscaled to resemble imaging of the same sample conducted at $0.5 \times 0.5 \times 0.5$ microns. This represents a huge increase in data size: going from 105 GB to 6.49 TB. Furthermore, the lower resolution stack is acquired in only 10 minutes (for a whole brain) compared to 16.5 hours to image the same brain at 0.5 micron cubic voxels. CACS therefore represents a huge time-saving if it can reliably provide artefact-free upscaling across the whole organ.

The work is impressive and is potentially of considerable interest to the community. The writing is generally clear. The videos are helpful. However, I have three broad concerns with the paper in its current form.

**** ONE ****

The relationship between the microscope itself and CACS is confusingly presented because the authors have chosen to describe their work as a single unit: a "content aware compressed sensing lightsheet microscope". For instance, in the Abstract on page 1 the authors state that they can image a mouse brain at $0.5 \mu\text{m}$ isovoxel resolution in 10 minutes. In fact, the sample was imaged $2 \times 2 \times 2$ micron voxels in 10 minutes and then up-scaled to 0.5 isovoxels post-hoc. This sort of thing persists throughout the paper, for instance on P. 9 there is a reference to "CACS-enabled Bessel sheet mode". To my mind, the microscope and the image analyses are independent: CACS is not an acquisition mode of the microscope, it's an optional analysis step. The microscope on its own is a useful advance and the analysis pipeline could potentially be applied to images acquired with other similar microscopes. The study would be more clearly presented and potentially of higher impact if more of a distinction were drawn between the hardware and the down-stream computational tools. I'm sure there will be readers who would want to apply CACS to their own data derived from a different SPIM design or even from serial-sectioning. If the paper would be more impactful if it could speak to these uses.

Techniques such as CACS inevitably have scenarios in which they work badly as well as scenarios where they work well. I don't currently have a feeling whether CACS behaves perfectly 100% of the time in the samples and resolutions shown or whether there are regions where it fails to a certain degree. For example, in supplementary video 2 the last slide shows large tile-like artefacts with variations in image quality. I don't understand what is happening there. Is this an artefact of the CACS? If so, how common is it? Please discuss: I believe there is only a brief mention of what is happening on page 5. On page 21 you state "If the signals are not dense and incoherent at the same time, CS allows the recovery of them from incomplete measurement". I don't have a feeling for what this means in practice. Are there examples for circumstances under which CACS fails? How likely is this to happen in real samples? What artefacts might your approach create in practice? Is there a trivial way of automatically flagging regions where the CACS does a bad job?

**** TWO ****

The current state of the whole organ 3-D imaging field is not well presented: the field is more advanced than the Introduction might lead a naïve reader to understand. For instance, there exist several solutions for 3-D imaging brains at sufficient resolution to count labelled somata in a reasonable time. The mesoSPIM and the LifeCanvas SmartSPIM can both do this in about about half an hour per channel. The field has no throughput problem at this resolution and the Introduction does not make this clear. Even serial-section 2-photon in practice tends not to be the bottleneck in experimental pipelines. The authors should also cite the FAST serial section spinning disk (Seiriki, 2019, <https://www.nature.com/articles/s41596-019-0148-4>) which images a whole mouse brain at 2.5 mm^3 in well under 3 hours without the hassle of clearing. The technique produce rich images of neurites.

Whole organ 3-D imaging is only cumberdously slow for situations where sub-cellular resolution is needed. The authors finally make this point at P.2 L.10 ("For example, both tiling confocal microscopy and sequential two-photon tomography can three-dimensionally image mouse brain at subcellular resolution, but do so at the expense of a very long acquisition time due to the slow laser-point-scanning.") and also at the top of P. 3. Before this there are somewhat vague or misleading statements such as:

P.1 L.3: "...thereby overcoming the throughput limit of current 3D microscopy implementations. "

P.2 L.5: "creating such a large-scale image dataset has posed significant challenges for current 3D light microscopy methods, which show relatively small optical throughputs"

P.3 L.4: "Gaussian and Bessel LSM systems still have limited optical throughput, which is far from adequate for obtaining high spatial resolution across a very large FOV"

In contrast, the advantages of the author's technique are under-sold in places. For instance, I am very impressed with the CACS-enhanced images showing dendrites and axons: this could be one of the most exciting uses for this approach. This is where $0.5 \mu\text{m}$ isovoxel resolution is really needed; for cell-counting it's unlikely to be needed in practice. Recent work by Winnubst et al. used serial-section 2-photon to image mouse brains at a voxel size of $0.3 \times 0.3 \times 1 \mu\text{m}$ and then trace neuronal projections of multiple cells per brain. Imaging took *one week* per brain. If CACS up-scaling to $0.5 \mu\text{m}$ isovoxels works as well as described, then imaging time could be cut down to a small fraction of that. This would be remarkable and of great interest to the field. However there isn't enough information to judge the technique's reliability with regard to brain-wide tracing of labelled axons. I think it's a mistake to relegate the bulk of the axonal tracing results to the supplementary figures and leave it poorly developed. I would suggest developing further the neuronal tracing presented in the manuscript. Can you reliably trace whole axonal trees as they project throughout the brain (it's hard to tell from supplementary Fig. 12 exactly what is going on)? You have a potential solution to the automated tracing problem on your doorstep: <http://dx.doi.org/10.1038/nmeth.3662> I would be super-excited if you could apply NeuroGPS-Tree to your data and trace potentially hundreds of cells per brain.

Regarding better developing the tracing: On Page 11 and supplementary figure 12. I find it hard to visualize which cells project to which areas. Are some incompletely traced? As well as tracing fibres in densely labeled tissue (e.g. Fig. 3), tracing thin axons over long distances is challenging but of great interest to the field. How readily can you do that? Are there artifacts from the CACS or from the imaging that might disrupt tracing? If so, how common are these? Your paper would be much stronger if you could provide compelling evidence that you have a system capable of creating datasets in which large numbers of single neurons can be accurately traced throughout the whole brain with an imaging time of under half a day (or even much faster).

**** THREE ****

I realise the authors' goal is probably not to develop a new microscopy community project; it is therefore not necessary to include detailed construction guides for the microscope. Nonetheless, there needs to be much more information in the supplementary material describing the mechanics, optics, design choices, pros and cons of the approach, how the microscope is run, etc, etc. There is currently not enough information for the technically-minded reader who might be contemplating building such a system.

- * I know some part numbers are listed in the methods section, but please provide a full parts list in table form. If available, please also provide for download a CAD model.
- * Given the advantages of the Bessel beam, why has it not been tried before for large samples? What problems did you have to solve to implement your solution?
- * What is the typical laser power at the sample?
- * Please provide more details on how the sample is mounted and how the chamber is constructed. More photographs and/or diagrams would help.
- * State which clearing techniques your design is compatible with. If there are limitations, please state whether they are potentially resolvable.
- * There is dual-sided excitation. On page 24 you state that at high resolutions you use just one side at a time. I assume this is because of the difficulty (shared with all SPIM systems) of perfectly aligning the two sheets in practice. If indeed this is the case, please discuss somewhere so the reader is clear what is going on (in my experience, most newcomers expect to use both sheets at the same time).
- * Supplementary Fig 1:
 - In the sentence "Finally, a Bessel plane illumination with widely tunable geometry (1-5 μm thickness, 1-20 mm width, 1-10 mm height)" I'm not certain I understand what "width" and "height" are. I think it would be helpful to have a plot showing how the axial length of the Bessel beam varies with beam thickness. i.e. what is the trade-off between the length of the beam and its thickness?
 - Please indicate on the diagram all critical distances. e.g. when components should be 1f apart.
 - State explicitly that the dashed box indicates the Gaussian path, which can be put into action with M7. It's not particularly clear as currently presented.

OTHER POINTS

* The introduction devotes substantial time to the problems associated with the small size of the Gaussian lightsheet waist. This nowadays is somewhat of a strawman and hardly worth mentioning as multiple groups have demonstrated the effectiveness of sweeping the waist. Shortly after discussion of these points the Introduction states that with a swept waist the "achieved axial resolution remains inadequate". This is too vague: inadequate for what? The mesoSPIM (one of the papers cited in relation to this claim) works in this way and shows it *is* possible to achieve resolution more than adequate for cell counting throughout the mouse brain. The authors should be more careful about wording and how they choose to represent their work in the context of the field as a whole.

* Page 2: The authors cite the Ragan serial section paper but should also cite at least one of Winnubst et al. "Reconstruction of 1,000 Projection Neurons Reveals New Cell Types and Organization of Long-Range Connectivity in the Mouse Brain. Cell 2019 and Economo et al. A New Platform for Brain-Wide Imaging and Reconstruction of Neurons. eLife 2016 These are more

appropriate than Helmchen & Denk.

* Page 4-7: starting here there are comparisons of the Bessel beam to a Gaussian beam. Since in this setup the waist of the Gaussian beam is not swept, I have a hard time comparing the results to the Bessel beam. In a realistic modern scenario the waist of the Gaussian beam would be swept. What, therefore, does "as compared with an over 20- μm thick Gaussian sheet" refer to? How thick is the sheet at the waist? What is the PSF like there? What are we seeing in supplementary figure 3? In supplementary figure 3 is the phrase "15 μm -thick regular Gaussian sheet". This implies Gaussian sheets are inevitably this thick, which of course is not so: it depends how they are made. You didn't optimise the Gaussian path (not swept waist and it seems from P.7 that the Gaussian path has a much lower excitation NA) so comparing the optimized Bessel beam to it doesn't seem to bring much to the table.

* Page 4: "(Fig. 1c) showed axial resolution and contrast superior to those from 3.2 \times epi-illumination and 3.2 \times Gaussian plane illumination (Fig. 1a, b)." Again, this just feels like a straw-man: of course your technique is better than epi. For the reasons above, I'm also not interested in the Gaussian sheet. What the reader cares about is how your microscope compares to a confocal: this is what most people would initially opt for if they wanted to trace dense neurites. Maybe this should be developed further (it's there in Fig 2).

* Supplementary Fig 9 (also reported on P.8): SNR does not seem like an appropriate descriptor for what you are calculating here, which seems more like a measure of contrast. e.g. I bet you could boost the "SNR" of image S9D by a factor of 10 simply by applying a high-pass filter.

* Supplementary Fig 10: This figure shows that confocal and TPE have a much higher photobleaching rate than SPIM. Confocal suffers from the "epi" problem, which explains what we see here. TPE excites in a selective plane and so does not suffer from this problem but still photobleaching is bad. Likely this is due to the higher photobleaching rate of TPE compared to 1p (Patterson & Piston, 2000). Maybe this should be mentioned and briefly discussed in the figure legend? I suspect a lot of readers will miss this point otherwise.

Page 8 and Fig 2: I know there is a citation, but I don't know what the "SPB limitation" is. Can you provide an intuitive grasp of it in one sentence? This would help clarify the statement "As shown in the data projected to a resolution-speed plane, the CACS Bessel sheet mode breaks the SPB limitation defined by the dashed resolution-speed curve. Thus, it can be considered to be a tool facilitating the combination of large FOV with isotropic high-resolution, which is difficult to meet using previous methods."

* Supplementary Fig. 3: please set $x=0$ at the peak of the curves. This would make the plots easier to read. The legend statement "yielding near-isotropic PSF" is difficult to support from the information presented in the figure. Maybe add a plot showing an x PSF cross-section?

* Supplementary Fig. 4: Since a major selling point of your microscope is the very fast acquisition speed at zoom 3.2 \times , please add data for this zoom to this figure. I think the line-like artefacts seen in a and b (top row) would probably go away if you interpolate the raw data over a finer grid before feeding it to the plot command.

* Supplementary Fig. 6: Can you please show at least some of these data in x - z also?

* Supplementary Fig. 11:

Although the images are in general very good, panels e and f show artefacts that look reminiscent of a very elongated PSF along the Z direction. These artefacts are multiple soma diameters in length, which surprises me a little because the 12.6 \times zoom PSF in Fig. SF4a looks better than that. Please comment on what is going on.

Page 10. In the discussion of Fig. 3 you state that you are using multiple views. It's a little unclear whether these are 90 degrees apart because you want to get a better PSF, or 180 degrees apart because you want to avoid imaging through the whole organ. Can you clarify, please? If it's the former, can you please discuss further in a supplementary figure?

Further, with a FOV of 4.2 mm, how many tiles are needed to cover the whole brain? Do I understand correctly that this tiling plus the two views is all imaged in 10 minutes?

* I understand from page 25 that it takes 5 hours to run CACS on a whole brain dataset going from 2 μm isovoxels to 0.5 μm isovoxels. You use 4 GPUs RTX2080ti to achieve this. How is data storage handled? Is SSD RAID needed to get this speed? Is the bottleneck the calculation or the IO? Given that the calculation is relatively quick but results in the creation of over 6 TB of data, please comment on the possibility of never generating a full up-sampled stack. e.g. I can imagine a 3-D viewer the dynamically up-scales the 2 μm iso-voxels on the fly. Similarly for axon tracing the stack could be up-scaled for tracing only. This would save a vast amount of storage space. I think providing more practical detail on the CACS implementation would be of interest to readers.

* Page 27 about the cell counting: what exactly do you mean by "This automatic counting procedure was also aided by human's crosscheck". Did you compare the automated results to ground-truth human counting? If so, how?

Questions relating to the registration

- Why do the authors use a non-validated registration process when validated alternatives exist (eg aMap)? This leaves the reader wondering how much fudging is required and their choice of registration software should be discussed. At the very least Github Elastix parameters and any relevant code associated with the registration. The reader might have some confidence if there were supplementary materials images to show how well this worked. Ten or so coronal sections with the overlaid Allen borders would be suitable. Supplementary Figure 14 doesn't quite show this. Examples of images that do can be found in Chon et al. (2019) doi: 10.1038/s41467-019-13057-w.

- I don't really understand your registration strategy. I don't follow the statement on page 26: "This pre-aligned brain was then resized into low-resolution and high-resolution groups." Forgive me if I misunderstand, but are you placing into the Atlas space the high res stacks in which you do the tracing? If so, why? It seems like a lot of work: it's easier to calculate the transform at low resolution (e.g. 10 or 25 $\mu\text{m}/\text{pix}$) and then transform only the traces to the Allen space.

- On page 25: "stitched images were iteratively registered for reconstructing a whole brain". What software did you use for this? If possible, please provide code and parameter files.

- On page 25 you state that "(2 μm voxel), which remains insufficient for single-cell/ neuron analysis". This is rather vague: what is it insufficient for? It's certainly a useful resolution in practice: our experience suggests that automated cell counting is perfectly feasible at 3 x 3 x 6 microns for labelled somata in cortex. Other authors have worked well with 1 x 1 x 50 microns (see Kim and Osten).

- On page 25 how were "... super-resolved patches ... automatically stitched together into a complete 3.5 trillion-voxel image"? Please supply more details in the supplementary materials if needed.

All of these comments would become void if you had chosen a published validated registration strategy that is accepted by the community.

MINOR POINTS

- * Page 1 Line 1: "Instant" probably the wrong word: it implies "real time". "Rapid" seems better.
- * Page 1 last line: "enabled" suggests these things could previously not be done. I would suggest "possible" instead.
- * Page 2 line 7: "in thick mammalian organs, severe light scattering and attenuation greatly limit the complete extraction of signals from deep tissues." You could probably remove this sentence. It doesn't contribute much.
- * Page 4. First sentence last paragraph: "but may still show inadequate resolutions". As earlier, "inadequate" for what? This is too vague.
- * Page 21: "Our Bessel plane illumination microscope (Supplementary Fig. 1, Video S1) sweeps a long circular Bessel beam in the y direction across the focal plane of the detection objective to create a large-scale scanned light-sheet and yield an image at a single z-plane within the specimen."
I don't think you mean "a long circular Bessel beam". The annulus is at the excitation objective back focal plane -- it's not the Bessel beam.

Reviewer #2:

Remarks to the Author:

The work presented by Fang et. al. has been demonstrated ultra-fast imaging on whole mouse brain by confocal slit Bessel light-sheet microscope. In addition, they used content-aware compressed-sensing (CACS) to improve the contrast and resolution based on a single acquisition. Confocal slit Bessel has been demonstrated and used by many groups, even in commercial product. Therefore, from the optical side, the reviewer didn't have questions about this part. Although the data is complete and very informative supplementary materials, this part is still not qualified for the submitted Journal because the optical method is quite familiar to this community. However, with the wide spread of computational microscopy that push typical light microscopy limit, the reviewer is happy to see a concrete implementation of compressive sensing (CS) toward resolving microscope data. But unfortunately, the reviewer didn't see much description about the concept of "content-aware" throughout the manuscript. I will there will be more paragraphs in the revision. And in the introduction part, it is quite short and focus on the light-sheet microscope. There should be some emphasis on the "CACS". The reviewer thinks the shining thing in the submitted paper should be CACS, not light-sheet microscope itself.

(1) lightsheet microscopy combined with tissue clearing could resolve whole mouse brain imaging. For the cleared sample preparation, the authors use organic phase to clear the sample, such as uDISCO, PEGASOS, which make the sample smaller and harden. That's the reason why they could do fast whole brain imaging by low resolution light-sheet imaging. Is this the case? Since the cleared sample often becomes soft and will suffer image registration problem by fast moving sample. The author should mention the sample requirement. The high resolution and imaging speed is mainly from the CACS reconstruction.

(2) Since the debut of sparse sampling concept back in 2004, there has been tons of image-based application for CS, but authors barely mentioned them in the introduction section (nor the discussion section). While the reviewer appreciate authors emphasize on why CS approach can be the next alternative/breakthrough for large scale sample imaging, this is not the pioneer work regarding CS itself. I would like to suggest authors to provide a brief intro that can differentiate the importance of content-aware (and differences with its CS origin).

(3) While the reviewer appreciates the reasoning in Supp Note 3 of why content-awareness (content-aware, CA) can mitigate overfitting issues and low single-to-noise ratio (SNR) issues in early CS trials, the reviewer believes that it is necessary to provide sufficient analytical reasoning

and proofs to support the claim for a high caliber journal. The only Supp description and its accompanied Supplementary citation only shed light on regularization terms, but not CA itself.

(4) The authors tried to claim that the whole organ imaging, but in the "Whole-brain CACS imaging procedure" section, where 3mm imaging depth was performed. This cannot cover the whole mouse brain. This has a concern that the what's the performance for CACS along with the imaging depth? In the manuscript, the demonstration for the comparison for CACS among other methods is done in the superficial area of the mouse brain, could the authors made a comparison with imaging depth as a parameter? or SNR ?

(5) The following issue is that, the reviewer appreciated that the authors made several awesome videos for the light-sheet scanning demonstration and brain image rendering, however, a video- z orthoslice for whole brain imaging for Thy-1 GFP neuron presented in the figure 2 or 3 will be much appreciated so that the readers could see how the resolution improved by CACS.

(6) The reviewer is happy to see that CS seems very practical in the lightsheet imaging; however, the content-aware by using synthetic PSF to iterate the reconstruction, there should be a training process which is missing in the manuscript; the following "influence" process to all tiles is already presented to some level in the manuscript. The "content aware" is mentioned page 23, equation (1-3), the reviewer wishes the authors address more on the CA, or cite some reference therein, for example, the reviewer found, Mathematical Problems in Engineering, Volume 2018, Article ID 7171352, 15 pages, should be informative to the authors.

(7) issues about coding, since "Code and Software Submission Checklist" is provided with the manuscript, the reviewer would like to suggest authors slightly reorganize their README file, requirements section (on GitHub) to follow the required contents section (the Checklist). While authors indeed stated their compile environment, CUDA 7.5 is hard to come by these days (perhaps, it is a restriction follow by MATLAB R2017?) if possible, please use a non-legacy CUDA version for the mex file. The reviewer is glad to see a proper inference section, which also complied with the Checklist. The reviewer would suggest authors to provide descriptions for trainings as well (though it seems to be optional and unchecked in the Checklist) to encourage future readers to apply their work in the field.

Reviewer #3:

Remarks to the Author:

In the manuscript the authors propose a fast Bessel beam light-sheet microscopy method to image large tissues and organs with isotropic subcellular resolution over a large field of view. Specifically, this is achieved by i) a slit scan confocal approach where the active pixels of the detection camera are synchronized with the sweeping bessel beam thus reducing unwanted contributions from the Bessel beam side lobes, and ii) a computational upsampling/deconvolution method termed CACS that restores high resolved images from low resolution acquisition by assuming sparsity of the signal. The authors demonstrate their method by tiled imaging whole mouse brain tissues at high isotropic resolution (1.5um) with order of magnitude higher throughput than classical (confocal/two photon) methods with comparable resolution.

Overall

In general I am very impressed with the results given in the manuscript and think that the proposed method can be a very valuable contribution to the field of whole tissue light-sheet imaging, where attaining isotropic subcellular resolution across millimeter sized field of view at high throughput is extremely challenging. I particularly like the different demonstrations of the methods value for quantitative downstream task such as nuclei counting and pre/postsynaptic

occupancy.

I do, however, still have some issues with i) the claimed optical resolution/efficiency of the confocal slit scan, and ii) the foundation/details of the CACS method, that when addressed, would make the paper more convincing in my opinion (see below).

Major

1. Characterisation of the overall resolution

In the abstract it is stated that "We demonstrate imaging [...] at subcellular resolution (0.5 μ m isovoxel)" -> that is confusing/misleading as the measured optical resolution in the manuscript is given as 1.5 μ m (after CACS).

In general the slit scan approach seems to already vastly increase the raw optical resolution even before CACS. However I am not entirely convinced by the measurements of this raw optical resolution and the comparison with the unsynchronised Bessel mode (Sup Fig 3/4), in particular:

- Sup Fig 3/4: what wavelength was used? What bead size?
- Sup Fig 3: The legend states that the larger scalebar is 50 μ m, thus the dotted line of the lineprofile should have length \sim 30 μ m. The axial profile in Sup Fig 3d however has length \sim 15 μ m? Additionally the FWHM of line-sync Bessel PSF seems to be \sim 1.8 μ m, which is smaller than the \sim 4 μ m given in the main text for the 3.2x Bessel beam?
- Sup Fig 3: Are all images normalized in the same way (e.g. background subtracted, or min/max scaled?)
- Sup Fig 3/4: The PSF of the unsynchronized (global shutter) Bessel sheet looks very strange. The PSF should be the product of the scanned Bessel sheet intensity and the detection PSF. So the axial FWHM should be bounded by the FWHM of the widefield PSF (which is smaller).

Altogether, I would like to see a more detailed description of the PSFs for each modality

2. CACS Method

Although being a fundamental contribution of the paper, the CACS method is in my opinion only superficially explained, specifically:

- How is the measurement matrix A defined? as circulant matrix?
- How is the synthetic PSF defined (e.g. which theoretical model)?
- What is the input and output (define size, subsampling factor...)
- It is stated that A is applied in Fourier domain - how are boundary artifacts handled (padding? tapering?)
- How is the subsampling/decimation process modelled?
- What specific method is used to solve Eq (1)? I don't quite understand the "steepest descent method" given in Eq (2): The right side is the L2 norm (and thus always positive)? and therefore X_{n+1} always increasing? Equally in Eq (3) it is not clear how X_{n+1} is computed (the left hand side is simply the L1 norm).
- How many steps are computed (how was convergence defined)?
- How are k_{line} , b_{line} selected? (Supp Note 3)
- What entropy definition is used? Simply that of the histogram? Which quantisation? (Supp Note 3)

Additionally Eq (1) is simple Basis pursuit denoising/Lasso. I am not really convinced that such a simple loss function would lead to such excellent deconvolution/upsampling results as shown in the paper in general situations without highly tuning all parameters (e.g. stepsize, entropy weighting factors k_{line} , b_{line}).

To address both issues, I therefore would like to see

- a quantitative evaluation of the performance of CACS on e.g. realistic, simulated data (with varying degree of sparsity) and varying subsampling factors
- a mathematically more detailed exposition of the CACS method including an explicit statement of all needed and computed quantities (e.g. subsampling factor, A , PSF, ...) and an explicit description of how to compute the updates $X_{\{n+1\}}$.

Minor

- How were the images for lower magnification upsampled before comparison with higher mag images for both the figures and the respective error map calculations? Judging from the images (e.g. the 2x images in Sup Fig 7 and Sup Fig 13) a simple nearest neighbour upscaling was used which can be misleading. I would like to see comparison against bilinear or bicubic upsampled images (which should have a lower SSIM/NRMSE).

- abstract: "improved by two-orders of magnitude" -> compared to what? Needs to be stated.

- Recently a light-sheet imaging modality (axially swept light-sheet microscopy) was reported that allows for imaging with comparable resolution/time (1mm^3 in 18min at $<1\mu\text{m}$ isotropic resolution), which should be appropriately mentioned:

Chakraborty et al. "Light-sheet microscopy of cleared tissues with isotropic, subcellular resolution." Nat. Methods (2019)

- typo: "abovementioned"

Point-by-point responses to reviewers' comments

Reviewer #1

Remarks to the Author:

Described is a wide field of view light sheet microscope that uses a scanned Bessel beam to selectively illuminate a thin plane in samples such as whole mouse organs. Unlike Gaussian light sheets, the Bessel beam (which is long and thin) does not require axially sweeping of the waist in order to obtain a sharp image throughout a large field of view. This allows for a substantial increase in acquisition speed. Furthermore, Bessel beams have "self healing" properties as they propagate: this makes them more immune to scattering than a Gaussian beam and so should improve image contrast in thick samples.

The authors augment their microscope using a computational pipeline: content-aware compressed sensing (CACS), which up-scales image resolution. CACS seems like a substantial improvement over conventional deconvolution techniques. A whole brain imaged at a pixel size of $2 \times 2 \times 2$ microns was upscaled to resemble imaging of the same sample conducted at $0.5 \times 0.5 \times 0.5$ microns. This represents a huge increase in data size: going from 105 GB to 6.49 TB. Furthermore, the lower resolution stack is acquired in only 10 minutes (for a whole brain) compared to 16.5 hours to image the same brain at 0.5 micron cubic voxels. CACS therefore represents a huge time-saving if it can reliably provide artefact-free upscaling across the whole organ.

The work is impressive and is potentially of considerable interest to the community. The writing is generally clear. The videos are helpful. However, I have three broad concerns with the paper in its current form.

**** ONE ****

The relationship between the microscope itself and CACS is confusingly presented because the authors have chosen to describe their work as a single unit: a "content aware compressed sensing lightsheet microscope". For instance, in the Abstract on page 1 the authors state that they can image a mouse brain at $0.5 \mu\text{m}$ isovoxel resolution in 10 minutes. In fact, the sample was imaged $2 \times 2 \times 2$ micron voxels in 10 minutes and then up-scaled to 0.5 isovoxels post-hoc. This sort of thing persists throughout the paper, for instance on P. 9 there is a reference to "CACS-enabled Bessel sheet mode". To my mind, the microscope and the image analyses are independent: CACS is not an acquisition mode of the microscope, it's an optional analysis step. The microscope on its own is a useful advance and the analysis pipeline could potentially be applied to images acquired with other similar microscopes. The study would be more clearly presented and potentially of higher impact if more of a distinction were drawn between the hardware and the down-stream computational tools. I'm sure there will be readers who would want to apply CACS to their own data derived from a different SPIM design or even from serial-sectioning. If the paper would be more impactful if it could speak to these uses.

1.1 We thank the reviewer for the professional advice. In this work, the CACS computation is combined with fast light-sheet microscopy to together maximize the imaging throughput. Besides light-sheet microscopy, CACS alone can work well with other 3D microscopy methods. In the

revised manuscript, we have added the CACS results of confocal and two-photon-excitation microscopy images (**new Supplementary Fig. 11**), which show similar resolution improvement and high recovery fidelity. In **Discussion** section of revised manuscript, we also mentioned the wide applicability of CACS as a separate image computation step, for increasing the throughput of other classic 3D imaging approaches.

Supplementary Fig. 11 | CACS computation for images obtained by confocal and TPE microscopes. A cortex region of mouse brain (Thy1-GFP-M) was imaged using confocal and TPE microscopes. 16x/0.8W objective lens was used for collecting the fluorescence signals. The confocal and TPE images acquired by rapid coarse scanning mode and slow fine scanning mode were defined as LR (voxel size $1.2 \times 1.2 \times 8 \mu\text{m}^3$) and HR (voxel size $0.31 \times 0.31 \times 2 \mu\text{m}^3$) images, respectively. **a-b**, Comparisons between LR, CACS and HR results of the x - y and x - z planes of the confocal cortex image. **c-d**, Comparisons between LR, CACS and HR results of the x - y and x - z planes of the TPE cortex image. The HR results were also regarded as ground truth to validate the resolution improvement and recovery accuracy of CACS results (error maps in 4th and 5th columns). **e-f**, SSIM and NRMSE values of LR (with 4x bicubic interpolation) and CACS results, calculated by using HR results as reference. Scale bar, 20 μm .

Techniques such as CACS inevitably have scenarios in which they work badly as well as scenarios where they work well. I don't currently have a feeling whether CACS behaves perfectly 100% of the time in the samples and resolutions shown or whether there are regions where it fails to a certain degree. For example, in supplementary video 2 the last slide shows large tile-like artefacts with

variations in image quality. I don't understand what is happening there. Is this an artefact of the CACS? If so, how common is it? Please discuss: I believe there is only a brief mention of what is happening on page 5. On page 21 you state "If the signals are not dense and incoherent at the same time, CS allows the recovery of them from incomplete measurement". I don't have a feeling for what this means in practice. Are there examples for circumstances under which CACS fails? How likely is this to happen in real samples? What artefacts might your approach create in practice? Is there a trivial way of automatically flagging regions where the CACS does a bad job?

1.2. We thank the reviewer for an interesting question about the limit of CACS computation. As previously reported, if the original signals are sparsely distributed in the image volumes (or can be sparsely represented by some given appropriate basis (e.g., in Fourier, or Wavelet space), CS allows the recovery of them from incomplete measurement^{1,2}, which is our case in large-FOV Bessel sheet imaging. Furthermore, for signals with different densities, regularization terms with different weighting can be used as constraints to optimize the recovery quality. The novelty of CACS is that it can automatically adjust the regularization for different unit regions based on the feedbacks of the image contents (signal density, degree of disorder), thereby achieving adaptive recovery of a large-scale 3D image that contains significant signal fluctuations. In **Supplementary Fig. 6-9** of revised manuscript, we have shown the recovery results for both point-like and line-like signals with various densities, and pointed out the minor artefacts arise in relatively dense areas. In general, when the regularization factor λ is less than 0.1, the signal will be highly dense and vulnerable to reconstructing errors. It should be noted that this is rarely the case in our whole-organ imaging, owing to the naturally incomplete measurement under large-FOV setup. Regarding the tile-like artefacts shown in **Video S2** (middle column) and mentioned by reviewer, they are just from the conventional CS without CA (fixed regularization, also see **Supplementary Note Fig. 1e**). The results of our CACS shown on the right column do not have this problem. In the revised manuscript, we have updated the **Results**, and **Methods** section to better clarify these points. We also provided more discussions on the performance of CACS recovery including its limitations (**Supplementary Fig. 6-9, 19, Supplementary Note 3, Note Fig. 1**)

**** TWO ****

The current state of the whole organ 3-D imaging field is not well presented: the field is more advanced than the Introduction might lead a naïve reader to understand. For instance, there exist several solutions for 3-D imaging brains at sufficient resolution to count labelled somata in a reasonable time. The mesoSPIM and the LifeCanvas SmartSPIM can both do this in about about half an hour per channel. The field has no throughput problem at this resolution and the Introduction does not make this clear. Even serial-section 2-photon in practice tends not to be the bottleneck in experimental pipelines. The authors should also cite the FAST serial section spinning disk (Seiriki, 2019, <https://www.nature.com/articles/s41596-019-0148-4>) which images a whole mouse brain at 2.5 mm³ in well under 3 hours without the hassle of clearing. The technique produce rich images of neurites.

1.3. We thank the reviewer for the professional advice. Axially scanned light-sheet microscopy (ASLM) and FAST spinning disk are both compelling 3D microscopy methods. We have provided more introduction to the waist-sweeping methods (**Ref. 24 and 25**) and FAST spinning disk (**Ref. 8**)

in the revised manuscript. As a reference point, mesoSPIM (**Ref. 24**), a newly reported whole-brain ASLM method, can image the whole mouse brain in 65 minutes, with a resolution of $1.6 \times 1.6 \times 2 \mu\text{m}^3$. The throughput is ~ 4 times lower than our Bessel sheet microscope (pure hardware, without CACS combined), which provides similar $2 \times 2 \times 2 \mu\text{m}^3$ resolution for whole mouse brain in 10 minutes. The main reason is that these waist-sweeping methods (mesoSPIM, SmartSPIM, ctASLM, etc) scan the waist of Gaussian light-sheet by zoomable lens such as electrical tunable lens (ETL), and the acquisition rate is not as high as galvo-scanner based approaches. While fast spinning disk also shows good results, its whole-brain imaging throughput is also ~ 4.5 times lower with achieving $0.7 \times 0.7 \times 5 \mu\text{m}^3$ resolution of in ~ 2.4 hours. Furthermore, we note that both ASLM and confocal microscopy have the issue of excessive out-of-waist (focus) excitation (nearly 100% intensity at out-of-focus regions), and thus cause nonnegligible photobleaching to samples (**Ref. 26**).

Whole organ 3-D imaging is only cumbersome slow for situations where sub-cellular resolution is needed. The authors finally make this point at P.2 L.10 ("For example, both tiling confocal microscopy and sequential two-photon tomography can three-dimensionally image mouse brain at subcellular resolution, but do so at the expense of a very long acquisition time due to the slow laser-point-scanning.") and also at the top of P. 3. Before this there are somewhat vague or misleading statements such as:

P.1 L.3: "...thereby overcoming the throughput limit of current 3D microscopy implementations. "

P.2 L.5: "creating such a large-scale image dataset has posed significant challenges for current 3D light microscopy methods, which show relatively small optical throughputs"

P.3 L.4: "Gaussian and Bessel LSM systems still have limited optical throughput, which is far from adequate for obtaining high spatial resolution across a very large FOV"

1.4. We thank the reviewer for the advices. While current microscope implementations can be either fast or high resolution, it is difficult for them to achieve both, owing to the insufficient imaging throughput which is known as the amount of spatial information provided per unit time (**Ref. 1**), and can be defined as (volumetric imaging speed) / (achievable spatial resolution) (**Ref. 31**). Compared to the conventional epi-illumination 3D microscopes, the throughput of light-sheet microscopy is much higher, but still insufficient for rapid (minute-level) and high-resolution (submicron) imaging of large samples (e.g., whole brain). The soul of our work is exactly about how to break this limit through combining the advanced optics with new computation pipeline: building a high-speed isotropic light-sheet microscope already with throughput advantage over classic histological imaging methods, and developing CACS computation to further boost the throughput by nearly two orders. In the revised manuscript, we further clarified the claims about throughput in the three places reviewer mentioned (P.1 L.23, P.2 L.7, P.3 L.9).

In contrast, the advantages of the author's technique are under-sold in places. For instance, I am very impressed with the CACS-enhanced images showing dendrites and axons: this could be one of the most exciting uses for this approach. This is where $0.5 \mu\text{m}$ isovoxel resolution is really needed; for cell-counting it's unlikely to be needed in practice. Recent work by Winnubst et al. used serial-section 2-photon to image mouse brains at a voxel size of $0.3 \times 0.3 \times 1 \mu\text{m}$ and then trace neuronal projections of multiple cells per brain. Imaging took *one week* per brain. If CACS up-scaling to $0.5 \mu\text{m}$ isovoxels works as well as described, then imaging time could be cut down to a small fraction of that. This would be remarkable and of great interest to the field. However there isn't

enough information to judge the technique's reliability with regard to brain-wide tracing of labelled axons. I think it's a mistake to relegate the bulk of the axonal tracing results to the supplementary figures and leave it poorly developed. I would suggest developing further the neuronal tracing presented in the manuscript. Can you reliably trace whole axonal trees as they project throughout the brain (it's hard to tell from supplementary Fig. 12 exactly what is going on)? You have a potential solution to the automated tracing problem on your doorstep: <http://dx.doi.org/10.1038/nmeth.3662> I would be super-excited if you could apply NeuroGPS-Tree to your data and trace potentially hundreds of cells per brain.

1.5. We are glad to know that our imaging results are impressive. As the reviewer has mentioned, higher-quality data indeed enables more applications. Because this work is mainly a report on a new imaging method, we previously just demonstrated relatively simple neuron segmentation / tracing in **Fig. 3** and **original Supplementary Fig. 12**. In response to reviewer's suggestion, now we have shown the automatic segmentation of dense neurons using the NeuroGPS-Tree (**Ref. 43**) (**revised Supplementary Fig. 15**), and further traced several long-distance projection neurons across multiple brain regions with revealing their annotated pathways (**revised Supplementary Fig. 16**).

Supplementary Fig. 15 | Tracing dense neurons in the cortex area of mouse brain (Thy1-GFP-M, 8 weeks). **a**, 3D CACS Bessel sheet image of a cortex area ($1 \times 1 \times 0.6 \text{ mm}^3$ volume) containing densely-packed pyramidal tract neurons. **b**, A neuronal tree with more than 200 neurons traced by applying the Neuro GPS-Tree¹ to our CACS Bessel sheet image. **c-j**, Detailed 3D views of 8 projection neurons initiated from this dense bundle. The results have validated that CACS image with achieving high volumetric resolution across large scale allows the automatic segmentation and tracing of dense projection neurons.

Regarding better developing the tracing: On Page 11 and supplementary figure 12. I find it hard to visualize which cells project to which areas. Are some incompletely traced? As well as tracing fibres in densely labeled tissue (e.g. Fig. 3), tracing thin axons over long distances is challenging but of great interest to the field. How readily can you do that? Are there artifacts from the CACS or from the imaging that might disrupt tracing? If so, how common are these? Your paper would be much stronger if you could provide compelling evidence that you have a system capable of creating datasets in which large numbers of single neurons can be accurately traced throughout the whole brain with an imaging time of under half a day (or even much faster).

1.6. We thank the reviewer for the advice and we think it is very constructive. First, as we state above, it has been widely demonstrated that the inaccuracy and artefacts in the reconstructed images of the whole brain is very minor. Even if it is encountered occasionally, the tracing would not be affected (**Supplementary Fig. 6-9**). In the **new Supplementary Fig. 15 and 16** of the revised manuscript, we have validated that based on our CACS dataset, single neurons across the brain can be precisely traced with their pathways clearly visualized.

**** THREE ****

I realise the authors' goal is probably not to develop a new microscopy community project; it is therefore not necessary to include detailed construction guides for the microscope. Nonetheless, there needs to be much more information in the supplementary material describing the mechanics, optics, design choices, pros and cons of the approach, how the microscope is run, etc, etc. There is currently not enough information for the technically-minded reader who might be contemplating building such a system.

* I know some part numbers are listed in the methods section, but please provide a full parts list in table form. If available, please also provide for download a CAD model.

1.7. We sincerely appreciate reviewer's advices, which help our work to be more complete. In the **revised Supplementary Fig. 1, 2 and Table S1**, we have provided the optical layout of the microscope, its photograph, full parts list, and CAD design of customized mechanical parts. Also, we have added a paragraph describing the system optics, how it runs, its pros and cons *etc* (**Methods**). We hope these additions will offer an easier time to the reader who might be interested in building such a system.

* Given the advantages of the Bessel beam, why has it not been tried before for large samples? What problems did you have to solve to implement your solution?

1.8. It's a good question. Bessel light-sheet microscopy has been previously reported, mainly for live cell imaging. Before the clearing techniques become established recently, the strong light scattering from thick tissues poses big challenge to the high-resolution imaging of large organs. In this case, the use of Gaussian light sheet is simpler and more cost-effective. (For example, the commercialized Ultramicroscope in **Ref. 15**). In addition, when imaging whole organs with low-magnification / large-DOF objective, it is especially important for us to reject the side lobe excitation of the Bessel beam by simultaneously sweeping the camera's rolling shutter, as discussed in the **Results and Supplementary Fig. 3, 4** of the manuscript. Therefore, the high isotropic resolution from our scanning Bessel sheet is also depended on the progress of sCMOS camera control in recent years. Actually, besides our large-scale Bessel sheet, ASLM techniques such as the aforementioned mesoSPIM and smartSPIM that sweep the waist of Gaussian beam, similarly benefit from the technical advances in tissue clearing and camera acquisition.

* What is the typical laser power at the sample?

1.9. The illumination power at the sample is ~ 4 mW for 488-nm laser and ~ 12 mW for 532-nm laser, respectively. The power on the focal plane excited by the central peak of Bessel beam is ~ 3 mW and 9 mW.

* Please provide more details on how the sample is mounted and how the chamber is constructed. More photographs and/or diagrams would help.

1.10. In the revised **Supplementary Fig. 2**, we have added the CAD designs and more photographs of the sample holder and chamber, to fully show how the sample is mounted and how the chamber is constructed.

Supplementary Fig. 2 | The mechanical design for sample holding. **a**, 3D layout of assembled sample mounting, translation and fluorescence detection parts. **b**, Detailed view of the sample mounting and 3D translation. A right-angle bracket (shown in **c**) was used to integrate the z translation stage with the xy stage, and a customized magnetic connector (shown in **d**) was designed to connect the 3D translation stage with the sample holder (shown in **e**). **e**, Detailed design of the sample holder. The cleared sample was mounted to the shaft of a water-proof motor (2, sealed in a resin box) through a FEP tube connector (1). The motor mounted with sample was then fixed with a rod-like magnet holder (3) using set screw. Such a sample holder containing brain sample and 1,2,3 parts can be readily attached to the magnetic connector (4, also shown in **d**) by using magnetic force. This design permits: 1. 3D translation and rotation of the sample in the solution. 2. convenient sample loading/unloading through the magnetic force. **f**, Photograph of the sample holder corresponding to the design layout shown in **e**. **g**, Photographs of an excised whole mouse brain before and after tissue clearing. **h**, Design of the solution container and its mounting part. **i**, Photograph showing that the scanned Bessel sheet is illuminating a transverse plane of the cleared mouse brain, and the detection objective is simultaneously collecting the fluorescence signals excited at the illuminated plane.

* State which clearing techniques your design is compatible with. If there are limitations, please state whether they are potentially resolvable.

1.11. We currently use dehydration-based clearing methods (uDISCO, fDISCO, *etc*), because the size shrinkage after clearing is suited for imaging of large organ, and the harden sample allows relatively easy mounting. Technically, the system can process various types of cleared samples, e.g., samples by Clarity, or Scale, after slightly modifying the holder design. In the revised manuscript, we have clarified the sample requirements and provided detailed description on the sample mounting (**Methods** section, P. 20, L. 22).

* There is dual-sided excitation. On page 24 you state that at high resolutions you use just one side at a time. I assume this is because of the difficulty (shared with all SPIM systems) of perfectly aligning the two sheets in practice. If indeed this is the case, please discuss somewhere so the reader is clear what is going on (in my experience, most newcomers expect to use both sheets at the same time).

1.12. There are two reasons for not using two sides at the same time: 1. The laser power can be increased without splitting, which allows shorter exposure time, higher frame rate and reduced acquisition time. 2. We have actually solved the problem of precise alignment on both sides. However, when the same area is illuminated from dual sides, the far-side illumination will induce more obvious tissue scattering, making the imaging effect even worse than single-side illumination. Therefore, in our experiment, the dual-side laser-sheets illuminate the sample not in a simultaneous but a sequential mode (e.g., left laser for 3 tiles in left column, and then right laser for another 3 tiles in right column), to increase the laser power and minimize the light scattering from the far-side illumination. Classic dual-side light-sheet microscopes, such as Ultramicroscope and mv-SPIM (**Ref. 15** and **37**), also use such sequential illumination mode. In the revised manuscript, we have clarified this point at **Methods** section.

* Supplementary Fig 1:

- In the sentence "Finally, a Bessel plane illumination with widely tunable geometry (1-5 μm thickness, 1-20 mm width, 1-10 mm height)" I'm not certain I understand what "width" and "height" are. I think it would be helpful to have a plot showing how the axial length of the Bessel beam varies with beam thickness. i.e. what is the trade-off between the length of the beam and its thickness?

1.13. We are sorry for the confusion brought to reviewer. In the **revised Supplementary Fig. 1**, we have added the geometry of Bessel sheet with its height, length, and thickness clearly indicated. We also added the plot of thickness versus diffraction length suggested by the reviewer.

Supplementary Fig. 1 | Dual-side, dual-mode light-sheet setup. **a**, A multi-wavelength laser was collimated and expanded to generate a Gaussian beam with 10 mm diameter. For each side, an axicon (AX) was used to generate a Bessel beam, which was further scanned into a plane by a galvo scanner (GM) and projected onto the sample using two groups of relay lenses (L1, L2, F-theta lenses, RO1-4). Dual Bessel sheets were precisely aligned by finely tuning the y-mirror of GM2. Finally, a Bessel plane illumination with widely tunable geometry (1-5 μm thickness, 1-20 mm FOV width, 1-10 mm height) was formed for rapid and high-axial-resolution imaging of large specimen. In our setup, a dual-side Gaussian light sheet (shown in the dashed box, switched by M1, 3, 5, 7) was also reserved for quick sample screening, as well as comparison with Bessel sheet. **b**, **c**, Geometry of scanning Bessel sheet and the correlation between its thickness (axial resolution) and diffraction length (width of FOV). **d**, Photograph of the microscope built on an optical bench.

- Please indicate on the diagram all critical distances. e.g. when components should be 1f apart.
- State explicitly that the dashed box indicates the Gaussian path, which can be put into action with M7. It's not particularly clear as currently presented.

1.14, 15. We appreciate reviewer's kind reminder. The descriptions reviewer suggested have been added to the revised Supplementary Fig. 1 and the caption.

OTHER POINTS

* The introduction devotes substantial time to the problems associated with the small size of the Gaussian lightsheet waist. This nowadays is somewhat of a strawman and hardly worth mentioning as multiple groups have demonstrated the effectiveness of sweeping the waist. Shortly after discussion of these points the Introduction states that with a swept waist the "achieved axial resolution remains inadequate". This is too vague: inadequate for what? The mesoSPIM (one of the papers cited in relation to this claim) works in this way and shows it is possible to achieve resolution more than adequate for cell counting throughout the mouse brain. The authors should be more careful about wording and how they choose to represent their work in the context of the field as a whole.

1.16. We thank the reviewer for the comments. As mentioned above, now we have provided more

introduction to ASLM methods, and clarified the statements in the revised manuscript. ASLM-type methods can also significantly improve the axial resolution and do have promising future. With further mitigating the limitation on acquisition rate, we believe it will become more powerful for whole-organ imaging.

* Page 2: The authors cite the Ragan serial section paper but should also cite at least one of Winnubst et al. "Reconstruction of 1,000 Projection Neurons Reveals New Cell Types and Organization of Long-Range Connectivity in the Mouse Brain. Cell 2019 and Economo et al. A New Platform for Brain-Wide Imaging and Reconstruction of Neurons. eLife 2016 These are more appropriate than Helmchen & Denk.

1.17. We have added the suggested references (Ref. 12, 13) in the revised manuscript.

* Page 4-7: starting here there are comparisons of the Bessel beam to a Gaussian beam. Since in this setup the waist of the Gaussian beam is not swept, I have a hard time comparing the results to the Bessel beam. In a realistic modern scenario the waist of the Gaussian beam would be swept. What, therefore, does "as compared with an over 20- μm thick Gaussian sheet" refer to? How thick is the sheet at the waist? What is the PSF like there? What are we seeing in supplementary figure 3? In supplementary figure 3 is the phrase "15 μm -thick regular Gaussian sheet". This implies Gaussian sheets are inevitably this thick, which of course is not so: it depends how they are made. You didn't optimise the Gaussian path (not swept waist and it seems from P.7 that the Gaussian path has a much lower excitation NA) so comparing the optimized Bessel beam to it doesn't seem to bring much to the table.

1.18. We thank the reviewer for the comments. While waist-sweeping methods recently emerged (e.g., mesoSPIM and ctASLM in 2019) for large-organ imaging, standard Gaussian light-sheet microscopy with anisotropic axial resolution ($\sim 10\text{-}20\ \mu\text{m}$ waist) is still the most well-known and commonly-used approach (SPIM, Ultramicroscope in Ref. 14, 15), also more accessible to us for comparison. Specially integrating an ALSM into our dual-side Bessel-sheet microscope for comparison is not practical, also not necessary. Our approach is mainly about the large-scale computational imaging at high resolution and throughput. In term of optics, it is sufficient to develop a Bessel light-sheet microscope with also isotropic resolution, and even higher original throughput. In the revised manuscript, we have stated that the comparison was made with static Gaussian light-sheet, and also pointed out that improved waist-sweeping light-sheet could achieve similar isotropic resolution like what Bessel sheet does.

* Page 4: "(Fig. 1c) showed axial resolution and contrast superior to those from $3.2\times$ epi-illumination and $3.2\times$ Gaussian plane illumination (Fig. 1a, b)." Again, this just feels like a straw-man: of course your technique is better than epi. For the reasons above, I'm also not interested in the Gaussian sheet. What the reader cares about is how your microscope compares to a confocal: this is what most people would initially opt for if they wanted to trace dense neurites. Maybe this should be developed further (it's there in Fig 2).

1.19. We appreciate reviewer's comments. The results in Fig. 1 showed a longitudinal comparison on how the serial light-sheet imaging techniques evolve from conventional wide-field microscopy. Fig. 2 further compared our approach with mainstream confocal and two-photon microscopes, including their SNR, resolution, speed, and overall throughput, to provide a comprehensive

performance study. In revised **Supplementary Fig. 12, 13, Supplementary Note Table 1**, we also validated the resolution and throughput advantages of CACS Bessel-sheet over those alternative approaches. Thus, we believe adequate performance comparisons with commonly-used 3D microscopes have been made in our manuscript now. Regarding the comparison on whole-brain imaging, it is impractical to image a cleared whole brain with a commercially-accessible confocal microscope, owing to the extremely long acquisition time, and lack of objective with enough working distance. Furthermore, we have added the introduction to state-of-the-art fast confocal microscope (**Ref. 8**) in the revised manuscript

* Supplementary Fig 9 (also reported on P.8): SNR does not seem like an appropriate descriptor for what you are calculating here, which seems more like a measure of contrast. e.g. I bet you could boost the "SNR" of image S9D by a factor of 10 simply by applying a high-pass filter.

1.20. While in **Supplementary Fig. 12**, we did calculate the image SNR rather than contrast according to an established algorithm (**Ref. 28**), we also agree with the reviewer that either the image contrast or SNR could be tuned by denoising techniques. In our demonstration, we compared the results of different modes under regular experimental settings/parameters listed in the **Supplementary Note Table 1**, without any additional denoising applied. In the revised caption of **Supplementary Fig. 12**, we further clarified this prerequisite for SNR comparison.

* Supplementary Fig 10: This figure shows that confocal and TPE have a much higher photobleaching rate than SPIM. Confocal suffers from the "epi" problem, which explains what we see here. TPE excites in a selective plane and so does not suffer from this problem but still photobleaching is bad. Likely this is due to the higher photobleaching rate of TPE compared to 1p (Patterson & Piston, 2000). Maybe this should be mentioned and briefly discussed in the figure legend? I suspect a lot of readers will miss this point otherwise.

1.21. We thank the reviewer for this very professional comment. TPE is based on non-linear two-photon absorption, to selectively excite the fluorophores at the vicinity of focal plane, and thus lead to relatively low out-of-focus bleaching. But at the same time, the high photon density required for nonlinear absorption will cause higher bleaching on the focal plane. In general, the bleaching rate of TPE imaging could be lower than confocal, but not much (**Ref. 1, 21**). Our results from the confocal and TPE modes of the same Nikon Ni-E microscope, also validated this well-accepted principle (**Fig. 2, Supplementary Note Table 1**). In the revised manuscript, we have added the explanation and citation on how the bleaching rates were measured (**Results** section).

Page 8 and Fig 2: I know there is a citation, but I don't know what the "SPB limitation" is. Can you provide an intuitive grasp of it in one sentence? This would help clarify the statement "As shown in the data projected to a resolution-speed plane, the CACS Bessel sheet mode breaks the SPB limitation defined by the dashed resolution-speed curve. Thus, it can be considered to be a tool facilitating the combination of large FOV with isotropic high-resolution, which is difficult to meet using previous methods."

1.22. We thank reviewer for the question. The spatial bandwidth product (SBP), herein defined as (volumetric imaging speed) / (achievable spatial resolution), actually represents the trade-off between imaging speed and resolution. In **Fig. 2k**, the colored dots on the dotted line have the same SBP, also meaning the same throughput. Without breaking the SBP limit, either increasing the

resolution or speed will make the colored dots move along the line, and cause the decrease of another one. Thus, the imaging throughput is not essentially improved. CACS Bessel-sheet imaging breaks this limitation so that its dot falls outside the line, indicating a throughput much higher than regular light-sheet imaging achieved. In the revised manuscript, we have explained the SBP limitation and its relationship with the throughput plot shown in **Fig. 2**.

* Supplementary Fig. 3: please set $x=0$ at the peak of the curves. This would make the plots easier to read. The legend statement "yielding near-isotropic PSF" is difficult to support from the information presented in the figure. Maybe add a plot showing an x PSF cross-section?

1.23. In the revised **Supplementary Fig. 3**, the bead intensity profiles along x axis have been plotted, to validate the near-isotropic PSF in our Bessel-sheet imaging.

Supplementary Fig. 3 | Comparison of different plane illumination modes. We used electronic confocal slit to eliminate side lobe excitation from Bessel beam and thereby increase axial resolution. When compared to a 15 μm -thick regular Gaussian sheet under $3.2\times/0.28$ detection, unsynchronized (high-speed dithering) Bessel sheet caused substantially elongated PSF due to the accumulated axial excitation by side lobes, which cannot be eliminated by the large depth-of-field detection objective. In contrast, synchronizing scanned Bessel sheet with electronic slit significantly reduced this side effect, yielding isotropic PSF with axial extent much shorter than that of Gaussian sheet. **a-c**, PSFs measured by Gaussian sheet, unsynchronized Bessel sheet, and synchronized Bessel sheet modes, as shown in **a-c**, respectively. **d, e**, Axial and lateral line profiles of the beads resolved by three methods. It should be noted that since the z -scan step size was set to 0.5 μm (oversampling) for these PSF measurements, the axial FWHM of PSF by synchronized Bessel sheet ($\sim 1.6 \mu\text{m}$) here corresponds to the native central-lobe thickness of Bessel sheet, which are thereby smaller than the axial FWHMs of neuron fibers ($\sim 4.5 \mu\text{m}$ in main text) measured using a 2- μm step size and even the lateral FWHM of PSF obtained by a 2- μm -pixel sampling. All plots were normalized in the same way. Scale bar, 50 μm (inset, 5 μm).

* Supplementary Fig. 4: Since a major selling point of your microscope is the very fast acquisition speed at zoom 3.2x, please add data for this zoom to this figure. I think the line-like artefacts seen in a and b (top row) would probably go away if you interpolate the raw data over a finer grid before feeding it to the plot command.

1.24. We thank the reviewer for the helpful advice. In the **revised Supplementary Fig. 4**, we have added the PSF results of synchronized / unsynchronized 3.2× Bessel sheet, and optimized the display with the line-like artefacts removed.

Supplementary Fig. 4 | Necessity for line synchronization under different magnification/NA setups.

It should be noted that the axial fluorescence contamination from side-lobe excitation is especially severe under low (3.2×)-to-middle (12.6×) magnification, which is our case, due to the extended depth-of-focus. We have validated this through imaging the fluorescence beads (~0.2 μm) under 3.2×/0.28 and 12.6×/0.5 detection using both synchronized and unsynchronized Bessel sheet modes, and comparing their axial performance with those by 20×/1.0 and 60×/1.2 detections. a-d, Axial performance of the PSFs (x-z planes) under 3.2×/0.28, 12.6×/0.5, 20×/1.0 and 60×/1.2 detection setups. The top and bottom rows show the synchronized and unsynchronized results, respectively. Scale bar, 1 μm. While all four setups showed deteriorated axial excitation by side-lobe excitation when line synchronization was not applied, the 3.2×/0.28 and 12.6×/0.5 setups were obviously more vulnerable to this side effect, owing to their relatively large depth-of-focus that received excessive axial signals excited by the side lobes. Therefore, the line synchronization is particularly necessary for our implementation. ↵

* Supplementary Fig. 6: Can you please show at least some of these data in x-z also?

1.25. We appreciate reviewer's suggestion. In the **revised Supplementary Fig. 7 and 9**, we have shown the results of neurons and cell nuclei in x-z planes, and also quantitatively analyzed the reconstruction accuracy.

Supplementary Fig. 7 | CACS computation for line-like neurons with different signal density at axial planes. x - z planes of the same four volumes shown in Fig. S6 were shown to compare the difference at axial planes. **a-d**, Comparisons between 3.2 \times , 3.2 \times CACS and 12.6 \times results of the four volumes in HB, CNU, HIP and CB brain regions, respectively. The axial planes (x - z planes) were visually inspected in the second to fourth columns. The error maps of 3.2 \times and 3.2 \times CACS results shown at fifth and sixth columns further validated the high fidelity of axial planes in CACS results. Scale bar, 20 μ m.

Supplementary Fig. 9 | CACS computation for point-like cell nuclei with different signal density at axial planes. a-d, Comparisons between 2x, 2x CACS and 8x results of the four volumes in PI-labelled hippocampus and isocortex, respectively. The axial planes (x-z planes) were visually inspected (2nd – 4th columns) and quantitatively evaluated (error maps in 5th and 6th columns), to validate the similarly significant improvement of axial performance in CACS results. Scale bar, 20 μm .

* Supplementary Fig. 11:

Although the images are in general very good, panels e and f show artefacts that look reminiscent of a very elongated PSF along the Z direction. These artefacts are multiple soma diameters in length, which surprises me a little because the 12.6x zoom PSF in Fig. SF4a looks better than that. Please comment on what is going on.

1.26. We thank the reviewer for the advice. We believe what the reviewers mentioned is the shadow artefacts near the neuron somata shown in Fig. 3e. This is because during synchronized Bessel sheet imaging (12.6x here), though the confocal slit will reject most of the side lobe excitation, but still not 100%. Meanwhile, in a few areas, the fluorescence intensity of soma can be tremendously higher than the fibers intensity (~100 times). Under this circumstance, even the artefacts caused by the weakened side-lobe excitation of soma might become observable. This issue is minor in our results and can be easily eliminated by a deconvolution or denoising. In the revised manuscript, we have optimized these images. In the revised manuscript, we have updated the **Supplementary Fig. 14** with artefact-free images.

Supplementary Fig. 14 | Scalable isotropic imaging of neurons in mouse brain. **a**, Dual-side, tunable Bessel sheet illumination combined with zoomable detection FOV from $1.26\times$ (~ 1 cm) to $12.6\times$ (~ 1 mm). **b**, Reconstructed coronal planes of four mouse brains that were rapidly screened using raw $1.26\times$ Bessel sheet mode. Each brain was imaged under 2 views for generating a multi-view-fused 3D reconstruction that contains 1000 image planes. The bottom row shows volume rendering of a whole brain (No.2) selected, owing to its best signal distribution among 4 candidates. **c**, MIPs in transverse (x - y ; top), coronal (x - z ; middle), and sagittal (y - z ; bottom) planes of the No.2 whole brain, showing the overall signal distributions. Then, higher-resolution imaging of any region of interest (ROI) was possible using the $12.6\times$ Bessel sheet mode. For example, we imaged three $\sim 8\times 10^{-3}$ mm³ regions of interest in the cortex, hippocampus, and cerebellum of No.2 brain at an imaging speed of ~ 0.01 mm³ s⁻¹ and an isotropic resolution of ~ 1.5 μ m (0.5 - μ m voxel). **d-f**, Three $\sim 8\times 10^{-3}$ mm³ volumes in cortex (yellow), hippocampus (green), and cerebellum (red) regions, selected from the coarse $1.26\times$ reconstruction, and further imaged by $12.6\times$ Bessel sheet mode, to reveal the various neuron types/structures (dense dendrites in **d**, pyramid neurons in **e**, astrocytes in **f**) at subcellular isotropic resolution. Scale bar, 20 μ m.

Page 10. In the discussion of Fig. 3 you state that you are using multiple views. It's a little unclear whether these are 90 degrees apart because you want to get a better PSF, or 180 degrees apart because you want to avoid imaging through the whole organ. Can you clarify, please? If it's the former, can you please discuss further in a supplementary figure?

Further, with a FOV of 4.2 mm, how many tiles are needed to cover the whole brain? Do I understand correctly that this tiling plus the two views is all imaged in 10 minutes?

1.27. We thank the reviewer for the professional questions. We image through the entire brain from 0- and 180-degree views. It's unnecessary to image the sample at 90-degree view because the PSF is already near-isotropic under each single view. With a 4.16×4.16 mm imaging FOV ($3.2\times$), we sequentially obtained 6 tiles (3 rows \times 2 columns) to cover the entire brain, and totally 12 tiles were acquired under two views. The 2-view acquisition, together with the sample rotation, movement, merely spend ~ 10 minutes. We have clarified these points in the **Results** and **Methods** section of revised manuscript (P. 11, L. 8 and P. 26, L.7).

* I understand from page 25 that it takes 5 hours to run CACS on a whole brain dataset going from

2 μm isovoxels to 0.5 μm isovoxels. You use 4 GPUs RTX2080ti to achieve this. How is data storage handled? Is SSD RAID needed to get this speed? Is the bottleneck the calculation or the IO? Given that the calculation is relatively quick but results in the creation of over 6 TB of data, please comment on the possibility of never generating a full up-sampled stack. e.g. I can imagine a 3-D viewer the dynamically up-scales the 2 μm iso-voxels on the fly. Similarly for axon tracing the stack could be up-scaled for tracing only. This would save a vast amount of storage space. I think providing more practical detail on the CACS implementation would be of interest to readers.

1.28. The reviewer's questions are very critical and suggestions are also very professional. The I/O time is indeed a big concern when processing, visualizing large-scale data. In our work, a 16T SSD RAID0 (SAMSUNG 2T \times 8) was created to guarantee the high-speed implementations of image stitching, registration, fusion, and CACS computation without much I/O delay. Exactly like what the reviewer has suggested, when visualizing the brain data using Imaris, the *ims* file provided hierarchical resolution when zooming in-and-out the image at different magnifications. An ordinary cellular resolution was enough for viewing the overall appearance of whole brain, while a subcellular resolution was provided when observing neuronal details in a small ROI. Such hierarchical-resolution display similar with running google map could greatly save the data storage and I/O time. For example, the *ims* file of a high-resolution whole brain occupied less than 2 TB data space, which was compared to original 6 TB for *tif* data. In response to reviewer's suggestion, we have provided more practical details about data processing and visualization in the revised manuscript (**Methods** section).

* Page 27 about the cell counting: what exactly do you mean by "This automatic counting procedure was also aided by human's crosscheck". Did you compare the automated results to ground-truth human counting? If so, how?

1.29. Yes, we used human counting to validate the accuracy of Imaris automatic counting. We first selected twenty volumes ($1 \times 1 \times 1 \text{mm}^3$) containing nuclei with different density and morphology for test. For each volume, we performed automatic counting, followed by a visual inspection. The averaged difference between the automatic and the manual counting results was verified to be less than 5%, a tolerable error rate. Then, the automatic cell counting was applied to the various regions of whole brain. In the revised manuscript, we have further clarified this point (**Methods** section).

Questions relating to the registration

- Why do the authors use a non-validated registration process when validated alternatives exist (eg aMap)? This leaves the reader wondering how much fudging is required and their choice of registration software should be discussed. At the very least Github Elastix parameters and any relevant code associated with the registration. The reader might have some confidence if there were supplementary materials images to show how well this worked. Ten or so coronal sections with the overlaid Allen borders would be suitable. Supplementary Figure 14 doesn't quite show this. Examples of images that do can be found in Chon et al. (2019) doi: 10.1038/s41467-019-13057-w.

- I don't really understand your registration strategy. I don't follow the statement on page 26: "This pre-aligned brain was then resized into low-resolution and high-resolution groups." Forgive me if I misunderstand, but are you placing into the Atlas space the high res stacks in which you do the tracing? If so, why? It seems like a lot of work: it's easier to calculate the transform at low resolution

(e.g. 10 or 25 $\mu\text{m}/\text{pix}$) and then transform only the traces to the Allen space.

1.30. We thank the reviewer for the questions on registration. We used a Elastix-based 3D image registration pipeline developed by ourselves (**Ref. 42**, similar to aMAP¹ but with higher registration accuracy) to transform our whole-brain data and register it with Allen's standard template. The obtained transformation correspondence was then applied to the Allen anatomical annotations to generate the transformed annotations for our Bessel brain. In fact, exactly like what the reviewer suggested, we obtained the transformation correspondence at a relatively low resolution to generate a Bessel brain map at 20 $\mu\text{m}/\text{pix}$ resolution. Meanwhile, Bessel brain image with native resolution was visualized in Imaris to facilitate the neuron and cell number analyses. Neuron tracing and cell counting were operated at high resolution and without the need of being registered to ABA. The segmented neurons / cells nuclei were then merged with the 3D brain map that was readily created at low resolution to obtain the pathway annotations. In the revised manuscript, we have stated how the registration was performed (**Methods** section), and added a figure showing the brain annotation/registration results, as the reviewer suggested (**new Supplementary Fig. 16**).

Supplementary Fig. 16 | 3D registration, region segmentation and neuron tracing of whole mouse brain (Thy1-GFP-M, 8 weeks). After imaging the whole brain using CACS Bessel sheet, we registered the whole-brain image with Allen's standard template using an accuracy-improved bi-channel image registration pipeline, which was developed by ourselves based on Elastix². Then we applied the Allen partition annotation file to divide and annotate the sub-regions in the registered mouse brain. **a1-a4**, Registered CACS brain (shown as different coronal planes) with annotation files applied. With the creation of a digital whole-brain atlas, we could also trace the pathways of long-distance projection neurons across the entire brain. **b1-b4**, 3D visualization of whole brain with showing the trajectories of long-distance projection neurons (**b1-b3**) and the brain sub-regions they pass through (**b4**). **c-g**, Five annotated projection neurons with pathway across different regions. **Ansiform lobule (AN)**, **Simple lobule (SIM)**, **Paramedian lobule (PRM)**, **Intermediate reticular nucleus (IRN)**, **Central lobule (CENT)**, **Declive (DEC)**, **Vestibular nuclei (VNC)**, **Uvula (UVU)** for **c**; **Ventral part (EPv)**, **Agranular insular area, ventral part (Alv6a)**, **Agranular insular area, posterior part (Alp6b)**, **Piriform area (PIR)** for **d**; **Ammon's horn (CA3)**, **lateral geniculate complex, dorsal part (LGd-sh)**, **Dentate gyrus, granule cell layer (DG-sg)**, **lateral posterior nucleus of the thalamus (LP)** for **e**; **Midbrain reticular nucleus (MRN)**, **Pons, behavior state related (P-sat)**, **Pons, motor related (P-mot)**, **Parvicellular reticular nucleus (PARN)**, **Vestibular nuclei (VNC)**, **Gigantocellular reticular nucleus (GRN)**, **Paragigantocellular reticular nucleus (PGRN)**, **Medullary reticular nucleus (MDRN)**, **Posterior amygdalar nucleus (Pa5)** for **f**; **Caudoputamen (CP)**, **External segment (GPe)**, **Ammon's horn (CA1)**, **Primary somatosensory area, upper limb (SSp-ul5)**, **Substantia nigra, reticular part (SNr)** for **g**.

- On page 25: "stitched images were iteratively registered for reconstructing a whole brain". What software did you use for this? If possible, please provide code and parameter files.

1.31. We used the open-source bead-based registration software developed by Preibisch *et al* (Ref. 40) to register-and-fuse two-view images into a complete 3D whole-brain image. The program files and other detailed information can be found at <https://www.nature.com/articles/nmeth0610-418>.

- On page 25 you state that "(2 μm voxel), which remains insufficient for single-cell/ neuron analysis". This is rather vague: what is it insufficient for? It's certainly a useful resolution in practice: our experience suggests that automated cell counting is perfectly feasible at 3 x 3 x 6 microns for labelled somata in cortex. Other authors have worked well with 1 x 1 x 50 microns (see Kim and Osten).

1.32. Reviewer's professional comments are appreciated. We also agree that the useful resolution in practice varies with the characteristics of signal to be analyzed. While the $3 \times 3 \times 6 \mu\text{m}^3$ voxel is indeed sufficient for the counting of non-dense signals like somata in cortex, a $2 \times 2 \times 2 \mu\text{m}^3$ voxel in raw $3.2 \times$ Bessel-sheet images might still show ambiguous dendrite details (**Fig. 2, 3**). Similarly, $1 \times 1 \times 50 \mu\text{m}^3$ voxel is only suited for counting very sparse signals (on-average axial spacing $>100 \mu\text{m}$), otherwise such a low sampling rate in z direction will cause inaccurate counting. In our application, since the PI-labeled nuclei signals are highly dense, the $3.25 \times 3.25 \times 4 \mu\text{m}^3$ voxel resolution in the raw $2 \times$ image remained insufficient for accurate cell counting (verified in **Fig. 4, Supplementary Fig. 17**), requiring the addition of CACS processing. In the revised manuscript, we have clarified the statement on the "insufficient resolution" (**Results** section, P. 8, L. 9). It should be noted that in practice, the CACS post-computation procedure is also flexible, users can opt into it or not according to the signal characteristics.

- On page 25 how were "... super-resolved patches ... automatically stitched together into a complete 3.5 trillion-voxel image"? Please supply more details in the supplementary materials if needed.

1.33. We used the open-source 3D tile stitching algorithm developed by Preibisch *et al* (**Ref. 39** in our manuscript) to stitch the image tiles. The program files and other detailed information can be found at <https://dl.acm.org/doi/abs/10.1093/bioinformatics/btp184>.

All of these comments would become void if you had chosen a published validated registration strategy that is accepted by the community.

1.34. As previously mentioned, our own brain mapping pipeline is similar to the established aMAP¹ and clear-MAP², and also developed based on open Elastix³. Meanwhile, it shows better registration/segmentation accuracy, especially for cleared brain data with noticeable shape deformation. The performance validation and other details of this method, termed BIRDS, have been revealed at our Biorxiv preprint (<https://doi.org/10.1101/2020.06.30.181255>).

MINOR POINTS

- * Page 1 Line 1: "Instant" probably the wrong word: it implies "real time". "Rapid" seems better.
- * Page 1 last line: "enabled" suggests these things could previously not be done. I would suggest "possible" instead.
- * Page 2 line 7: "in thick mammalian organs, severe light scattering and attenuation greatly limit the complete extraction of signals from deep tissues." You could probably remove this sentence. It doesn't contribute much.
- * Page 4. First sentence last paragraph: "but may still show inadequate resolutions". As earlier, "inadequate" for what? This is too vague.
- * Page 21: "Our Bessel plane illumination microscope (Supplementary Fig. 1, Video S1) sweeps a long circular Bessel beam in the y direction across the focal plane of the detection objective to create a large-scale scanned light-sheet and yield an image at a single z -plane within the specimen."

I don't think you mean "a long circular Bessel beam". The annulus is at the excitation objective back focal plane -- it's not the Bessel beam.

1.35-39. We thank the reviewer for pointing out these minor issues. In the revised manuscript, we have updated all the places mentioned by the reviewer, including the introduction to waist-sweeping methods (P. 1, L.16, 29, P. 2, L. 9, 28 and P. 4, L.14)

Reviewer #2

Remarks to the Author:

The work presented by Fang et. al. has been demonstrated ultra-fast imaging on whole mouse brain by confocal slit Bessel light-sheet microscope. In addition, they used content-aware compressed-sensing (CACS) to improve the contrast and resolution based on a single acquisition. Confocal slit Bessel has been demonstrated and used by many groups, even in commercial product. Therefore, from the optical side, the reviewer didn't have questions about this part. Although the data is complete and very informative supplementary materials, this part is still not qualified for the submitted Journal because the optical method is quite familiar to this community. However, with the wide spread of computational microscopy that push typical light microscopy limit, the reviewer is happy to see a concrete implementation of compressive sensing (CS) toward resolving microscope data. But unfortunately, the reviewer didn't see much description about the concept of "content-aware" throughout the manuscript. I will there will be more paragraphs in the revision. And in the introduction part, it is quite short and focus on the light-sheet microscope. There should be some emphasis on the "CACS". The reviewer thinks the shining thing in the submitted paper should be CACS, not light-sheet microscope itself.

2.1. We are happy to see that the reviewer appreciates the idea of CACS computation. In the revised manuscript, we have added a lot more descriptions on how the content-aware CS works and more validations on its performance. Meanwhile, we also agree that since synchronized Bessel light-sheet microscopy has been previously reported on small scale (**Ref. 28, 29**), the concept of side-lobe rejection with confocal slit is not so novel. Aside from the idea itself, in practice, we also need to make a lot of optimizations on the optical and mechanical design, to realize high-throughput Bessel imaging at whole-organ scale. Thus, we believe that the construction of such a zoomable isotropic organ imaging system, and its wide biomedical applications are still unique. This is quite similar with the care of MesoSPIM technique (**Ref. 24**, Nature Methods, 2019). While the principle of MesoSPIM is just from previously reported axially-scanned light-sheet microscopy (ASLM) (**Ref. 23**), it achieves superior imaging of whole brain through building a highly-specialized microscope. Furthermore, the development of new CACS in conjunction with large-scale Bessel sheet microscope together achieve ultra-high-throughput organ imaging. The overall performance of this hardware + software hybrid strategy is novel and compelling.

(1) lightsheet microscopy combined with tissue clearing could resolve whole mouse brain imaging. For the cleared sample preparation, the authors use organic phase to clear the sample, such as uDISCO, PEGASOS, which make the sample smaller and harden. That's the reason why they could do fast whole brain imaging by low resolution light-sheet imaging. Is this the case? Since the cleared sample often becomes soft and will suffer image registration problem by fast moving sample. The author should mention the sample requirement. The high resolution and imaging speed is mainly

from the CACS reconstruction.

2.2. We currently use dehydration-based clearing methods (uDISCO, fDISCO, *etc*), because the size shrinkage after clearing is suited for imaging of large organ, and the harden sample allows relatively easy mounting. Technically, the system can process various types of cleared samples, e.g., samples by Clarity, or Scale, after slightly modifying the holder design (e.g., replacing the screw clamping with a customized chamber for soften samples). For soften samples with size expansion, we can use lower-magnification detection objective to acquire raw data with equivalent resolution because the spatial distance between microstructures also increases in the expanded tissues and the required resolution for distinguishing the same structures becomes lower. Therefore, the $3.2\times$ detection ($2\text{-}\mu\text{m}$ voxel) used for a PEGASOS brain can be switched to $2\times$ ($3.25\text{-}\mu\text{m}$ voxel) for a CLARITY brain, with maintaining similar imaging speed as well as effective resolution. In the revised manuscript, we have further clarified the sample requirements and provided detailed description on the sample mounting (**Methods** section, P. 23, L. 22).

(2) Since the debut of sparse sampling concept back in 2004, there has been tons of image-based application for CS, but authors barely mentioned them in the introduction section (nor the discussion section). While the reviewer appreciate authors emphasize on why CS approach can be the next alternative/breakthrough for large scale sample imaging, this is not the pioneer work regarding CS itself. I would like to suggest authors to provide a brief intro that can differentiate the importance of content-aware (and differences with its CS origin).

2.3. We thank reviewer for the comments. We previously discussed the development, application and limitation of conventional CS in the **Results** and **Discussion** sections. In response to the reviewer's advice, we have enriched them and put them to the **Introduction** section of the revised manuscript (P. 3, L. 17).

(3) While the reviewer appreciates the reasoning in Supp Note 3 of why content-awareness (content-aware, CA) can mitigate overfitting issues and low single-to-noise ratio (SNR) issues in early CS trials, the reviewer believes that it is necessary to provide sufficient analytical reasoning and proofs to support the claim for a high caliber journal. The only Supp description and its accompanied Supplementary citation only shed light on regularization terms, but not CA itself.

2.4. We appreciate reviewer's helpful advice. CACS can dynamically adjust the balance between overfitting and underfitting to generate optimal outputs, through giving the regularization term an appropriate weighting factor λ , which is calculated based on the feedbacks of the image contents (signal density indicator α , signal disorder indicator β). Thus, the entire computation procedure is driven by the automatic content-awareness of the image, to achieve an optimized recovery superior to conventional CS. In the revised manuscript, we have added detailed descriptions on how to calculated the image content indicators and how they are linked with the regularization. We also experimentally validated that this content-aware CS recovery is necessary for processing large-scale image that contains variation of signal characteristics (**new Supplementary Note 3, revised Note Fig. 1**).

(4) The authors tried to claim that the whole organ imaging, but in the "Whole-brain CACS imaging procedure" section, where 3mm imaging depth was performed. This cannot cover the whole mouse brain. This has a concern that the what's the performance for CACS along with the imaging depth?

In the manuscript, the demonstration for the comparison for CACS among other methods is done in the superficial area of the mouse brain, could the authors made a comparison with imaging depth as a parameter? or SNR?

2.5. We thank reviewer for the question. The thickness of entire brain is ~5 mm. Although the brain is cleared, minor light scattering still exists in the deep tissues (**Fig. 3b1, 2**). Therefore, we applied a dual-view (0° and 180°) imaging strategy combined with a registration-and-fusion to mitigate this limitation (**Methods**, P. 26, L. 10). Under each view, 1500 images per each tile were acquired with a depth range from 0-3 mm (covering ~70% depth of the entire brain with the deepest 30% discarded). Then, the two-view 3D images were registered and fused into a whole brain image (2500 images, **Fig. 3a, b**) using a multi-view fusion program (**Ref. 40**). The fused whole brain shows uniform signal quality across its depth, which benefits the follow-up CACS reconstruction as well. In the revised manuscript, we have clarified the image acquisition process in **Methods** section (P. 26, L. 6), and added the SNR comparison of superficial layer/deep layer in **revised Fig. 3b** to validate the uniform CACS recovery quality across entire depth of brain.

Fig. 3 | Whole-brain mapping pipeline. **a**, Flow chart of whole-brain data acquisition and processing, which includes: **i**, rapid image acquisition by 3.2× Bessel sheet, with total 12 tiles under 2 views acquired in ~10 minutes. **ii**, tile stitching for each view followed by 2-view weighted fusion, to reconstruct a scattering-reduced whole brain at low resolution (2- μ m voxel). The full implementation of this step would be not necessary if the sample is optimally cleared or imaged under lower magnification. **iii**, CACS computation to recover a digital whole brain with improved resolution (0.5- μ m voxel) and SNR. **iv**, quantitative analyses, such as brain region segmentation, neuron tracing and cell counting, based on high-quality whole brain reconstruction. **b**, MIPs of reconstructed coronal plane (x - z , 150- μ m thickness) in whole brain by 0°, 180° single view Bessel sheet, 2-view fusion and CACS computation. The magnified views of two small regions selected from $z = 100$ and $z = 4500$ μ m depth further show the effect of 2-view fusion and CACS. Scale bar, 50 μ m. **c**, 3D visualization of a whole digital mouse brain (left, ~400 mm^3 , 3-trillion voxels) reconstructed from the 3.2× Bessel-CACS results, whose raw data was rapidly acquired in ~10 minutes. Three magnified volumes from the cortex (blue), cerebellum (red) and midbrain (orange) regions by raw 3.2× Bessel sheet (top row), 3.2× Bessel-CACS (middle row), and 12.6× Bessel sheet (bottom row) modes are compared to show the super-resolution capability of CACS procedure. Scale bars, 1 mm for whole brain visualization, 50 μ m for ROIs. **d**, SSIM and NRMSE values of 12 brain sub-regions ($100 \times 100 \times 100$ μm^3) images by raw Bessel sheet and Bessel-CACS. The quantitative analyses verify a high recovery fidelity of CACS results as compared to the high-resolution ground truths. **e**, Tracing of a single pyramidal neuron under three modes. The same neuron imaged by three modes was segmented and traced using Imaris. The results clearly show the improved segmentation accuracy (white arrows) brought by CACS. Scale bar, 50 μ m.

(5) The following issue is that, the reviewer appreciated that the authors made several awesome videos for the light-sheet scanning demonstration and brain image rendering, however, a video- z orthoslice for whole brain imaging for Thy-1 GFP neuron presented in the figure 2 or 3 will be much appreciated so that the readers could see how the resolution improved by CACS.

2.6. We appreciated the reviewer's advice, which is beneficial to the quality improvement of our

work. We have modified the **Supplementary Video S3** with also exhibiting the coronal planes (x - z orthoslices) of whole brain, to demonstrate the similar resolution enhancement by CACS.

(6) The reviewer is happy to see that CS seems very practical in the lightsheet imaging; however, the content-aware by using synthetic PSF to iterate the reconstruction, there should be a training process which is missing in the manuscript; the following “influence” process to all tiles is already presented to some level in the manuscript. The “content aware” is mentioned page 23, equation (1-3), the reviewer wishes the authors address more on the CA, or cite some reference therein, for example, the reviewer found, *Mathematical Problems in Engineering*, Volume 2018, Article ID 7171352, 15 pages, should be informative to the authors.

2.7. We thank the reviewer for the comments. Unlike convolutional neural network, our CACS is unsupervised and does not require a prior training process. The prior knowledge we need for CACS are the signal density and entropy of measurement data, which can be instantly calculated. In the revised manuscript, we have clarified this point (P. 3, L. 24). Also, as mentioned above, we have made detailed discussion on CA process (**Methods** section, **Supplementary Note 3**), with more references about CA cited as well (**Ref. 35, 36**).

(7) issues about coding, since “Code and Software Submission Checklist” is provided with the manuscript, the reviewer would like to suggest authors slightly reorganize their README file, requirements section (on GitHub) to follow the required contents section (the Checklist). While authors indeed stated their compile environment, CUDA 7.5 is hard to come by these days (perhaps, it is a restriction follow by MATLAB R2017?) if possible, please use a non-legacy CUDA version for the mex file. The reviewer is glad to see a proper inference section, which also complied with the Checklist. The reviewer would suggest authors to provide descriptions for trainings as well (though it seems to be optional and unchecked in the Checklist) to encourage future readers to apply their work in the field.

2.8. We appreciate reviewer’s advices on code implementation. We have optimized the code and reorganized the README file for better sharing our approach with future readers. The updated CACS code now supports CUDA 10.1, and we have tested it with Matlab R2020a and CUDA10.1.

Reviewer #3

Remarks to the Author:

In the manuscript the authors propose a fast Bessel beam light-sheet microscopy method to image large tissues and organs with isotropic subcellular resolution over a large field of view. Specifically, this is achieved by i) a slit scan confocal approach where the active pixels of the detection camera are synchronized with the sweeping Bessel beam thus reducing unwanted contributions from the Bessel beam side lobes, and ii) a computational upsampling/deconvolution method termed CACS that restores high resolved images from low resolution acquisition by assuming sparsity of the signal. The authors demonstrate their method by tiled imaging whole mouse brain tissues at high isotropic resolution (1.5 μ m) with order of magnitude higher throughput than classical (confocal/two photon) methods with comparable resolution.

Overall

In general I am very impressed with the results given in the manuscript and think that the proposed method can be a very valuable contribution to the field of whole tissue light-sheet imaging, where attaining isotropic subcellular resolution across millimeter sized field of view at high throughput is extremely challenging. I particularly like the different demonstrations of the methods value for quantitative downstream task such as nuclei counting and pre/postsynaptic occupancy.

I do, however, still have some issues with i) the claimed optical resolution/efficiency of the confocal slit scan, and ii) the foundation/details of the CACS method, that when addressed, would make the paper more convincing in my opinion (see below).

Major

1. Characterisation of the overall resolution

In the abstract it is stated that "We demonstrate imaging [...] at subcellular resolution (0.5 μ m isovoxel)" -> that is confusing/misleading as the measured optical resolution in the manuscript is given as 1.5 μ m (after CACS).

3.1. We apologize for the confusion brought to reviewer. The CACS-enhanced image has an isotropic voxel size of 0.5 μ m. According to Nyquist sampling law, theoretically the highest resolution achievable is \sim 1 μ m. The stated 1.5- μ m resolution after CACS is based on experimental measurements of a number of fine neuron fibers. In the revised manuscript, we have better clarify this point (P. 8, L. 23).

In general the slit scan approach seems to already vastly increase the raw optical resolution even before CACS. However I am not entirely convinced by the measurements of this raw optical resolution and the comparison with the unsynchronised Bessel mode (Sup Fig 3/4), in particular:

- Sup Fig 3/4: what wavelength was used? What bead size?

3.2. The laser wavelength was 488 nm. We used sub-diffraction beads with size \sim 0.5 μ m to

characterize the $3.2\times$ light-sheet imaging (Supplementary Fig. 3). Then $\sim 0.2\text{-}\mu\text{m}$ beads were used to compare how the confocal slit synchronization affected the axial performance of Bessel sheet illumination under different detection magnifications (**Supplementary Fig. 4**).

- Sup Fig 3: The legend states that the larger scalebar is $50\mu\text{m}$, thus the dotted line of the lineprofile should have length $\sim 30\mu\text{m}$. The axial profile in Sup Fig 3d however has length $\sim 15\mu\text{m}$? Additionally the FWHM of line-sync Bessel PSF seems to be $\sim 1.8\mu\text{m}$, which is smaller than the $\sim 4\mu\text{m}$ given in the main text for the $3.2x$ Bessel beam?

3.3. We appreciate reviewer's professional question. It's noted that since the z -scan step size was set to $0.5\ \mu\text{m}$ (oversampling) for the PSF measurements, the axial FWHM of PSF ($\sim 1.6\ \mu\text{m}$) by synchronized Bessel sheet here corresponds to the native central-lobe thickness of Bessel sheet ($1/e^2$ thickness $\sim 2.7\ \mu\text{m}$), which are thereby smaller than the axial FWHMs of neuron fibers ($\sim 4\ \mu\text{m}$ in main text) measured using a $2\text{-}\mu\text{m}$ step size. We have clarified this issue in the revised manuscript (**Results and Supplementary Fig. 3**).

- Sup Fig 3: Are all images normalized in the same way (e.g. background subtracted, or min/max scaled?)

3.4. We normalized the intensity plot instead of the images. All data points on the plot were normalized by $(x-x_{\min})/(x_{\max}-x_{\min})$.

- Sup Fig 3/4: The PSF of the unsynchronized (global shutter) Bessel sheet looks very strange. The PSF should be the product of the scanned Bessel sheet intensity and the detection PSF. So the axial FWHM should be bounded by the FWHM of the widefield PSF (which is smaller).

3.5. We thank reviewer for the comment. It is well known that the cross-sectional profile of the excitation sheet contains broad tails because of the combined influence of the side lobes. Given the fact that the drastic increase of depth-of-focus is proportional to the square of the decrease in NA, the axial excitation from these side lobes is more likely to deteriorate the fluorescence detection in our low-to-middle magnification/NA setup ($3.2\times$ and $12.6\times$, **revised Supplementary Fig. 4**), causing long line-like artefacts along z axis. In contrast, the axial excitation under high magnification/ NA detection is much better, exactly owing to the reason reviewer mentioned.

Altogether, I would like to see a more detailed description of the PSFs for each modality

2. CACS Method

Although being a fundamental contribution of the paper, the CACS method is in my opinion only superficially explained, specifically:

- How is the measurement matrix A defined? as circulant matrix?

3.6. First we sincerely thank the reviewer for a series of detailed questions (from 6-16). Answering them also substantially helps us to clarify our approach and improve quality of manuscript.

The measurement matrix A represents the unit response of the whole imaging system in Fourier space. Multiplying A with high-resolution Fourier image x to be solved (Fourier form of high-resolution target X) is equivalent to the imaging process in spatial domain, also known as the

convolution of PSF and target X . Therefore, the 3D measurement matrix A , also termed compressed matrix, is the Fourier transformation of the system PSF. To address reviewer's comment, in the revised manuscript, we have provided detailed description on how to generate the measurement matrix A from the synthetic PSF (**new Supplementary Note 4 and Note Fig. 2**).

- How is the synthetic PSF defined (e.g. which theoretical model)?

3.7. We generate the synthetic PSF for CACS computation according to the simulations of Bessel-sheet imaging process, separated into blurring of detection objective, the involvement of Bessel-sheet illumination, elimination of side-lobe excitation, and digitalization of camera acquisition. In the revised manuscript, we have provided detailed description on how to generate the synthetic PSF (**new Supplementary Note 4 and Note Fig. 2**).

- What is the input and output (define size, subsampling factor...)

3.8. Input is the low-resolution image y_i , which is the Fourier form of the raw measurement Y_i . Output is the high-resolution Fourier image x_i solved by the CACS. Parameters like voxel size, interpolation factor are initialized in the program. In the revised manuscript, we have included a detailed description on the implementation of iterative CACS solving procedure (**Supplementary Note 5**).

- It is stated that A is applied in Fourier domain - how are boundary artifacts handled (padding? tapering?)

3.9. The cropped volumes calculated by CACS indeed have boundary artefacts. So when we crop the large image volume to a number of small tiles for computation, we make the borders of adjacent small volumes overlapped with each other. The discontinued artefacts arise in these padding boundary areas are discarded before stitching the recovered tiles back into a large-size output.

- How is the subsampling/decimation process modelled?

3.10. During the imaging (degradation) process, fluorescence signals are blurred by the optical system and then pixelated by the sCMOS sensor. The subsampling/decimation process was modeled by a 3D Gaussian-Bessel blurring, followed by a pixel-averaged downsampling, using a point source as target. As we mentioned above, we have detailed the modeling process in the **new Supplementary Note 4 and Note Fig. 2**.

- What specific method is used to solve Eq (1)? I don't quite understand the "steepest descent method" given in Eq (2): The right side is the l_2 norm (and thus always positive)? and therefore X_{n+1} always increasing? Equally in Eq (3) it is not clear how X_{n+1} is computed (the left hand side is simply the l_1 norm).

3.11. We thank reviewer for the detailed questions. Now we have elucidated the iterative solving process of the equation in the **new Supplementary Note 5**. The "steepest descent method" is quoted from **Ref. 48**. We did not introduce it too much because it is not the main claim in our work. A brief introduction to how the iteration runs is also included in the **new Supplementary Note 5**, and more details can be found in **Ref. 48**.

- How many steps are computed (how was convergence defined)?

3.12. We have introduced the iteration parameters in **new Supplementary Note 5**. A *rel_gap* is defined as the quotient of duality gap η and dual function G . When the parameter *rel_gap* is less than the threshold ϵ , the equation is considered to reach the convergence condition and the iteration stops. A too-high ϵ will cause insufficient improvement and a too-low ϵ will suffer from additional time cost. In our implementation, ϵ is set to be 10^{-2} , a reasonable value validated via prior tests on several trial images.

- How are *k_line*, *b_line* selected? (Supp Note 3)

3.13. The labelled signals shown in this work are classified as point-like nuclei and line-like neurons, so we also divided the signals into these two types and tested our CACS by those selected ROIs which mainly contain cell bodies and neuron fibers. After a large number of tests, the optimal solutions for k and b of multiple sets of images were obtained. Since we found that signal entropy E and defined regularized parameter β have a linear correlation ($r > 0.9$), an equation $\beta_i = k \cdot E_i + b$ could be created to automatically determine the β from calculated E , with k and b coefficients for point- and line-like signals being obtained via a curve fitting. In the **revised Supplementary Note 3** and **Note Fig. 1**, a detailed description on this point has been included.

- What entropy definition is used? Simply that of the histogram? Which quantisation? (Supp Note 3)

3.14. We apologize for a bit vague description on the regularization terms and image entropy in the previous version manuscript. An established function *entropy* in Matlab, was used to calculate the image entropy which indicates the degree of signal disorder in the measurement data. In the **revised Supplementary Note 3**, we have stated this point with citation added.

Additionally Eq (1) is simple Basis pursuit denoising/Lasso. I am not really convinced that such a simple loss function would lead to such excellent deconvolution/upsampling results as shown in the paper in general situations without highly tuning all parameters (e.g. stepsize, entropy weighting factors *k_line*, *b_line*).

3.15. In conventional CS implementation, the recovery of relatively dense signals tends to have over-fitting issue caused by excessive initial constraints, thus showing poor improvement. On the contrary, the recovery of relatively sparse signals require more constraints to prevent too sharp artefacts⁴. Since the signals across a large FOV varies dramatically, conventional CS implementation with fixed regularization unsurprisingly causes obvious artefacts and signal-loss in the final stitched result (**Supplementary Note Fig. 1e**). The content-aware regularization factor λ_i introduced in our CACS procedure can reasonably judge the signal characteristics and dynamically balance the results between these two extremes, thereby recovering relatively accurate signals with substantial resolution improvement (**Supplementary Note Fig. 1f, g**). For example, as the calculated λ_i being large, also meaning sparse and ordered signals presented in raw image, the algorithm correspondingly has large weighting from the raw image and tends to preserve more existing real information rather than inference of new information. In the revised manuscript, we have elucidated how to reasonably extract the signal characteristics (entropy, density) and link them with optimized regularization via determining an appropriated weighting factor (**revised Supplementary Note 3**). Aside from claiming the advantages of our CACS processing, we have also noted that the CS computation has its applicable range (which fits our case, **Methods** section,

P. 24, L. 5), and shows performance ceiling (minor artefacts in highly dense regions in **Supplementary Fig. 6-9**, failure case when up-sampling factor is too high in **Supplementary Fig. 19**)

To address both issues, I therefore would like to see

- a quantitative evaluation of the performance of CACS on e.g. realistic, simulated data (with varying degree of sparsity) and varying subsampling factors
- a mathematically more detailed exposition of the CACS method including an explicit statement of all needed and computed quantities (e.g. subsampling factor, A, PSF, ...) and an explicit description of how to compute the updates X_{n+1} .

3.16. We thank reviewer for the advice. Through responding to abovementioned questions, we have made all the additions reviewer suggested in the revised manuscript. Please see the updated **Introduction** (P. 3, L. 17-26), **Methods** (P. 23, L. 2-28) and **Supplementary Information** (P. 26-30). Regarding the effect of CACS, we have shown the restoration effects of different sparsity and different types of signals (point/line) in **Supplementary Figs. 6-9** and **Supplementary Note Fig. 1**. The **Supplementary Fig. 10** has shown the effect of CACS at high magnification (12.6×). Regarding different subsampling factors, now we have shown the restoration effects of different sub factors in the **revised Supplementary Fig. 19**.

Minor

- How were the images for lower magnification upsampled before comparison with higher mag images for both the figures and the respective error map calculations? Judging from the images (e.g. the 2x images in Sup Fig 7 and Sup Fig 13) a simple nearest neighbour upscaling was used which can be misleading. I would like to see comparison against bilinear or bicubic upsampled images (which should have a lower SSIM/NRMSE).

3.17. We thank reviewer for the questions. According to previous work⁵, we applied Bicubic up-sampling instead of a simple nearest neighbor upscaling. We have clarified this point in the revised figure captions.

- abstract: "improved by two-orders of magnitude" -> compared to what? Needs to be stated.

3.18. We appreciate the reviewer's advice. We have made a more detail description in the **revised Abstract** section. The performance is compared to conventional 3D microscopy implementations, such as confocal and two-photon-excitation microscopes.

- Recently a light-sheet imaging modality (axially swept light-sheet microscopy) was reported that allows for imaging with comparable resolution/time (1mm³ in 18min at <1um isotropic resolution), which should be appropriately mentioned:

Chakraborty et al. "Light-sheet microscopy of cleared tissues with isotropic, subcellular resolution." Nat. Methods (2019)

3.19. We thank reviewer for the information. We have cited this paper (**Ref. 25**) and added comments in **revised Introduction**. At the same time, we also adjusted the statement on our own approach.

- typo: "abovementioned"

3.20. We have corrected this typo, and carefully read through the revised manuscript to wipe off other typos.

Response References

- 1 Niedworok, C. J. et al. aMAP is a validated pipeline for registration and segmentation of high-resolution mouse brain data. *Nature Communications* 7, 11879, doi:10.1038/ncomms11879 (2016).
- 2 Renier, N. et al. Mapping of Brain Activity by Automated Volume Analysis of Immediate Early Genes. *Cell* 165, 1789-1802, doi:10.1016/j.cell.2016.05.007 (2016).
- 3 Klein, S., Staring, M., Murphy, K., Viergever, M. A. & Pluim, J. P. W. elastix: A Toolbox for Intensity-Based Medical Image Registration. *Medical Imaging IEEE Transactions on* 29, P.196-205 (2010).
- 4 Cherkassky, V. & Mulier, F. *Learning from data : concepts, theory, and methods.* (2007).
- 5 Weigert, M. et al. Content-aware image restoration: pushing the limits of fluorescence microscopy. *Nature Methods* (2018).

Reviewers' Comments:

Reviewer #1:

Remarks to the Author:

The authors have addressed most of the concerns raised to my satisfaction. The manuscript is certainly much stronger now. I have the following two points to raise based on the revisions.

1.

The relationship between CACS and the Bessel light sheet.

It's very nice to see that CACS can be applied to other imaging modalities. Thanks for providing these data. Is it the case that the content-aware nature of CACS means no tweaking of parameters is needed to apply the algorithm to these different modalities? Adding a sentence about this to the figure legend would be helpful.

In my original review I also raised under this point the conflating of CACS with imaging. The abstract still states that the authors *image* a whole brain at 0.5 μm isovoxel resolution in 5 to 10 minutes. This is not the case: the imaging is conducted at substantially lower resolution then upscaled.

2.

The authors reply that cell counting was manually cross-checked. However, in supplementary fig 7 the "counting accuracy" seems to be with respect to the 8x image not a human rater.

3.

The added text near the start of the Introduction is misleading. It states that "...current 3D microscopy methods... ..have a limited imaging depth (up to a few hundred microns)...". That's obviously not true: tissue clearing has been around for some time.

4.

At the end of the first page of the Introduction: "Meso-SPIM" should be "mesoSPIM".

5.

Legend of Figure 3, last sentence: "The results clearly show the improved segmentation accuracy (white arrows) brought by CACS". The reader has no way of judging tracing accuracy based on 3E. I believe that CACS does indeed help but either you need to provide a different figure to support that statement or you need to modify the statement.

6.

It would be nice if the authors provided a GitHub (or similar) link to code rather than stating "available on request".

Reviewer #2:

Remarks to the Author:

In general, the latest revision acknowledged my primary concern with compressive sensing and its modeling. It would be our pleasure to support its publication.

While authors provide significant revisions in their supplementary materials and rebuttal responses, main text remains roughly the same. As briefly suggested by other reviewers, under current manuscript structure, they would undersell their CACS concept. We would like to suggest authors to port and restructure majority of their supplementary materials, especially the rationale of CACS applicability and its validity among datasets, into the main text.

Bi-Chang

Reviewer #3:

Remarks to the Author:

In the revision the authors have improved the paper considerably, added proper references, clarified most issues and especially extensively expanded the exposition of the CACS algorithm.

There are still two issues from the original round of review that I felt were not properly addressed in the response/revision which I regard to be important to be fixed in the final version:

1) Resolution claim

The resolution claim of 0.5 μ m in the abstract

"subcellular resolution (0.5- μ m isovoxel)"

remained unchanged, despite not being supported by the paper (which shows 1.5 μ m).

Although the authors write that

"In the revised manuscript, we have better clarify this point"

I would expect this to be changed in the abstract too.

2) Upsampling method used for comparison

In response to one concern, the authors state that for Supp Fig 7 (now Supp Fig 8) and Supp Fig 13 they included now comparison against bicubic upsampled images instead of nearest neighbour upsampling.

Although the caption of both figures suggests that change e.g. in Sup Fig. 8 "SSIM and NRMSE values of LR 2 \times results (with 4 \times bicubic interpolation)", both figures are still identical to the old ones and the SSIM/NRMSE values haven't changed although they should have. This is a bit confusing.

Minor:

- "confocal and TPE methods, respectively (Fig. 2k)" -> shouldn't that be Fig 2i?

- In the CACS description it is stated

"Since the intensity of signal in mouse brain has a large dynamic range [...] it can be sparsely distributed and recovered in the Fourier domain."

But in Eq (1), the sparsity constraint is put only on the signal, e.g. in the CACS method the sparsity assumption is about the image domain solely. This should be explicitly stated.

Point-by-point response to reviewers

Reviewer #1 (Remarks to the Author):

The authors have addressed most of the concerns raised to my satisfaction. The manuscript is certainly much stronger now. I have the following two points to raise based on the revisions.

1.

The relationship between CACS and the Bessel light sheet. It's very nice to see that CACS can be applied to other imaging modalities. Thanks for providing these data. Is it the case that the content-aware nature of CACS means no tweaking of parameters is needed to apply the algorithm to these different modalities? Adding a sentence about this to the figure legend would be helpful.

1.1.1 The reviewer is right. When applying CACS to different microscopy modalities, we only need to change parameters related with the imaging conditions (e.g., objective NA, voxel size) to generate appropriated PSFs for initializing the computation. We thank the reviewer for the helpful suggestion. We have clarified this point in the revised legend of Supplementary Fig. 11.

In my original review I also raised under this point the conflating of CACS with imaging. The abstract still states that the authors *image* a whole brain at 0.5- μm isovoxel resolution in 5 to 10 minutes. This is not the case: the imaging is conducted at substantially lower resolution then upscaled.

1.1.2 We appreciate the reviewer's comment. In the revised manuscript, we have changed the statement as "We demonstrate the imaging of whole brain ($\sim 400 \text{ mm}^3$), entire gastrocnemius and tibialis muscles ($\sim 200 \text{ mm}^3$) of mouse at ultra-high throughput of 5~10 minutes per sample and post-improved subcellular resolution (0.5- μm iso-voxel size)"

2.

The authors reply that cell counting was manually cross-checked. However, in supplementary fig 7 the "counting accuracy" seems to be with respect to the 8x image not a human rater.

1.2 In Supplementary Fig. S8, 9, we used 8 \times images as reference to quantify the reconstruction accuracy of 2 \times CACS results. In Supplementary Fig. S17, we compared the automatic cell counting results of 2 \times LR, 8 \times HR and 2 \times CACS using Imaris software. In this case, the visual inspection was introduced to verify the high accuracy of the automatic counting procedure. We have clarified this point in the revised legend of Supplementary Fig. 17.

3.

The added text near the start of the Introduction is misleading. It states that "...current 3D microscopy methods... ..have a limited imaging depth (up to a few hundred microns)...". That's obviously not true: tissue clearing has been around for some time.

1.3 The reviewer is right, tissue clearing has been around for a few years. Here we just want to emphasize that before the tissue clearing being combined with the advanced microscopy techniques, whole-tissue-scale 3D imaging, such as STPT and tiling confocal, relies on the physical sectioning of the tissue because the imaging depth is limited to up to a few hundreds of microns. To avoid the possible confusion to readers, now we have changed the phrase "current 3D microscopy methods"

into “conventional 3D microscopy methods” in the revised manuscript

4.

At the end of the first page of the Introduction: “Meso-SPIM” should be “mesoSPIM”.

1.4 We have corrected this typo in the revised manuscript.

5.

Legend of Figure 3, last sentence: “The results clearly show the improved segmentation accuracy (white arrows) brought by CACS”. The reader has no way of judging tracing accuracy based on 3E. I believe that CACS does indeed help but either you need to provide a different figure to support that statement or you need to modify the statement.

1.5 In response to the reviewer’s suggestion, we have modified statement as “It’s shown that CACS result enables the identification of apparently more neuron sub-structures.”

6.

It would be nice if the authors provided a GitHub (or similar) link to code rather than stating "available on request".

1.6 We thank the reviewer for the suggestion. The Github link of the code has been provided in the Code Availability section of the revised manuscript.

Reviewer #2 (Remarks to the Author):

In general, the latest revision acknowledged my primary concern with compressive sensing and its modeling. It would be our pleasure to support its publication.

While authors provide significant revisions in their supplementary materials and rebuttal responses, main text remains roughly the same. As briefly suggested by other reviewers, under current manuscript structure, they would undersell their CACS concept. We would like to suggest authors to port and restructure majority of their supplementary materials, especially the rationale of CACS applicability and its validity among datasets, into the main text.

2.1 We appreciate the reviewer’s helpful advice. Through the last-round revision, we have discussed a lot more on the principle of CACS and demonstrated its performance for different types of biological samples. In this revised manuscript, we have further included more CACS results and related discussions in the main text (revised Fig. 1 and Results section).

Reviewer #3 (Remarks to the Author):

In the revision the authors have improved the paper considerably, added proper references, clarified most issues and especially extensively expanded the exposition of the CACS algorithm.

There are still two issues from the original round of review that I felt were not properly addressed in the response/revision which I regard to be important to be fixed in the final version:

1) Resolution claim

The resolution claim of 0.5 μ m in the abstract "subcellular resolution (0.5- μ m isovoxel)" remained unchanged, despite not being supported by the paper (which shows 1.5 μ m). Although the authors write that "In the revised manuscript, we have better clarify this point" I would expect this to be changed in the abstract too.

3.1 We apologize for the confusion brought to the reviewer. The size of the voxel in our reconstructed brain image is 0.5 μ m. According to the Nyquist sampling theorem, the theoretical image resolution is over 1 μ m in this case. In practice, the averaged FWHMs of measured thin dendrite fibers was \sim 1.5 μ m, which was regarded as the achieved resolution in the brain imaging. In the revised manuscript, we have clearly clarified the 1.5- μ m resolution and 0.5- μ m voxel size in the abstract.

2) Upsampling method used for comparison

In response to one concern, the authors state that for Supp Fig 7 (now Supp Fig 8) and Supp Fig 13 they included now comparison against bicubic upsampled images instead of nearest neighbour upsampling.

Although the caption of both figures suggests that change e.g. in Sup Fig. 8 "SSIM and NRMSE values of LR 2 \times results (with 4 \times bicubic interpolation)", both figures are still identical to the old ones and the SSIM/NRMSE values haven't changed although they should have. This is a bit confusing.

3.2 The upsampling method we used in this work is bicubic interpolation all the time. In the last-round revision, we just added the statements to clarify the upsampling method. Therefore, the calculated SSIM/NRMSE values was not changed.

Minor:

- "confocal and TPE methods, respectively (Fig. 2k)" -> shouldn't that be Fig 2i?

3.3 We thank the reviewer for pointing out this mistake. We have corrected it in the revised manuscript.

- In the CACS description it is stated "Since the intensity of signal in mouse brain has a large dynamic range [...] it can be sparsely distributed and recovered in the Fourier domain." But in Eq (1), the sparsity constraint is put only on the signal, e.g. in the CACS method the sparsity assumption is about the image domain solely. This should be explicitly stated.

3.4 We thank the reviewer for the comment. In the revised manuscript, we have used the uppercase and lowercase letters to distinguish the signals in the spatial domain or the Fourier domain. The uppercase letters X/Y represent the signals in the spatial domain, and the lowercase letters x/y represent the signals in the Fourier domain. Therefore, the sparsity constraint is put on the signals in Fourier domain. We have explicitly stated this in the revised manuscript.